# Transcription-associated topoisomerase 2α (TOP2A) activity is a major effector of cytotoxicity induced by G-quadruplex ligands

Madeleine Bossaert[1,2†], Angélique Pipier[1,2†], Jean-Francois Riou[3], Céline Noirot[4], Linh-Trang Nguyên[1], Remy-Felix Serre[5], Olivier Bouchez[5], Eric Defrancq[6], Patrick Calsou[1,2‡§*], Sébastien Britton[1,2‡§*], Dennis Gomez[1,2‡*]

[1]Institut de Pharmacologie et Biologie Structurale, IPBS, Université de Toulouse, CNRS, UPS, Toulouse, France; [2]Equipe Labellisée Ligue Contre le Cancer 2018, Toulouse, France; [3]Structure et Instabilité des Génomes, Muséum National d'Histoire Naturelle, CNRS, INSERM, Paris, France; [4]INRAE, UR 875, Unité de Mathématique et Informatique Appliquées, Genotoul Bioinfo, Castanet-Tolosan, France; [5]INRAE, US 1426, GeT-PlaGe, Genotoul, Castanet-Tolosan, France; [6]Département de Chimie Moléculaire, UMR CNRS 5250, Université Grenoble Alpes, Grenoble, France

*For correspondence:
patrick.calsou@ipbs.fr (PC);
sebastien.britton@ipbs.fr (SéB);
dennis.gomez@ipbs.fr (DG)

†These authors contributed equally to this work
‡These authors also contributed equally to this work
§Corresponding author

Competing interests: The authors declare that no competing interests exist.

**Abstract** G-quadruplexes (G4) are non-canonical DNA structures found in the genome of most species including human. Small molecules stabilizing these structures, called G4 ligands, have been identified and, for some of them, shown to induce cytotoxic DNA double-strand breaks. Through the use of an unbiased genetic approach, we identify here topoisomerase 2α (TOP2A) as a major effector of cytotoxicity induced by two clastogenic G4 ligands, pyridostatin and CX-5461, the latter molecule currently undergoing phase I/II clinical trials in oncology. We show that both TOP2 activity and transcription account for DNA break production following G4 ligand treatments. In contrast, clastogenic activity of these G4 ligands is countered by topoisomerase 1 (TOP1), which limits co-transcriptional G4 formation, and by factors promoting transcriptional elongation. Altogether our results support that clastogenic G4 ligands act as DNA structure-driven TOP2 poisons at transcribed regions bearing G4 structures.

## Introduction

In recent years, evidence has accumulated to indicate that transcription is a major source of genomic instability (*Aguilera, 2002*; *Gaillard et al., 2013*; *Gómez-González and Aguilera, 2019*). Transcription-dependent DNA double-stranded breaks (DSBs) are mainly associated with RNA-polymerase (RNA-Pol) arrests provoked by different non-exclusive factors including DNA torsional stress, inhibition of transcription elongation, and formation of secondary structures, such as G-quadruplexes (G4) and R-loops (*Chedin and Benham, 2020*; *Crossley et al., 2019*; *García-Muse and Aguilera, 2019*; *Kotsantis et al., 2020*; *Miglietta et al., 2020*). G4 are four-stranded secondary structures formed at guanine-rich tracts (*Burge et al., 2006*). Present throughout the human genome (*Chambers et al., 2015*; *Hänsel-Hertsch et al., 2016*; *Hänsel-Hertsch et al., 2018*), G4 have been associated with spontaneous DNA breaks, hotspots for chromosomal translocations and several human syndromes (*Hänsel-Hertsch et al., 2017*; *Murat and Balasubramanian, 2014*; *Maizels, 2015*; *Puget et al., 2019*). In proliferating cells, G4 act as replication fork barriers, provoking fork collapses, the activation of the DNA damage response and the induction of replication-dependent DSBs (*Lerner and*

*Sale, 2019*). In addition, increasing evidence also indicates a significant impact of G4 structures on genomic stability through transcription-dependent processes (*Kotsantis et al., 2020*; *Miglietta et al., 2020*; *De Magis et al., 2019*; *Rodriguez et al., 2012*; *van Wietmarschen et al., 2018*; *Kim, 2019*).

G4 mapping in the human genome shows a significant enrichment of these structures within promoter and 5′ UTR regions of highly transcribed genes, and several *in vitro* and cellular studies show that the stabilization of G4 structures by small compounds, G4 ligands, generally represses transcription of genes containing G-rich tracts (*Chambers et al., 2015*; *Hänsel-Hertsch et al., 2016*; *Hänsel-Hertsch et al., 2017*). During transcription, while G4 structures located on template DNA could act as physical barriers blocking RNA-Pol II progression, the formation of G4 on the opposite strand could promote and stabilize secondary structures that block transcription elongation (*Gómez-González and Aguilera, 2019*; *Kotsantis et al., 2020*; *Miglietta et al., 2020*). Genome-wide analyses of G4 motifs in human cells indicate that these structures are highly correlated with RNA-Pol II pausing sites and R-loop-forming regions, two different factors promoting RNA-Pol II arrests and transcription-dependent DNA breaks (*Puget et al., 2019*; *Chen et al., 2017*; *Eddy et al., 2011*). Additionally, the formation of highly stable DNA secondary structures, such as G4, has been shown to promote the formation of DNA topoisomerase 2 (TOP2)-mediated DNA breaks (*Szlachta et al., 2020*). Interestingly, recent studies demonstrate a major contribution of TOP2 activity in the generation of DSBs in highly transcribed genes (*Canela et al., 2019*; *Gittens et al., 2019*; *Gothe et al., 2019*). Moreover, several studies demonstrated that DNA topoisomerase 1 (TOP1) and TOP2 recognize and preferentially cleave DNA at regions forming stable secondary structures (*Froelich-Ammon et al., 1994*; *Jonstrup et al., 2008*; *Mills et al., 2018*; *West and Austin, 1999*). In eukaryotic cells, TOP1 and TOP2 activities are required to resolve topological stresses resulting from DNA transactions (for a review, see *Pommier et al., 2016*). These enzymes relax topological constraints through the formation of transient single-stranded (TOP1) or double-stranded DNA breaks (TOP2), in which the enzymes are covalently linked to the DNA backbone. In humans, TOP2 activity is supported by two isoenzymes, TOP2α (TOP2A) and TOP2β (TOP2B), that are encoded by two different genes. TOP2A plays key roles in DNA replication and chromosome segregation, while TOP2B is mainly associated with transcription (*Pommier et al., 2016*; *Nitiss, 2009a*; *Madabhushi, 2018*). TOP2 are poisoned by small molecules that trap the transient TOP2-DNA complex, also known as 'cleavage complex' (TOP2cc), during the enzyme catalytic cycle (*Deweese and Osheroff, 2009*; *Nitiss, 2009b*; *Pommier et al., 2010*). The repair of TOP2cc requires a sequential process consisting in the removal of TOP2 protein from DNA through TOP2 proteolysis (*Fan et al., 2008*; *Mao et al., 2001*; *Zhang et al., 2006*) or nucleolytic degradation of the associated DNA (*Aparicio et al., 2016*; *Nakamura et al., 2010*) and the repair of the resulting DSB by non-homologous end joining (NHEJ) (*Gómez-Herreros et al., 2013*) or homologous recombination (HR), respectively (*Ashour et al., 2015*).

*Bruno et al., 2020* and *Olivieri et al., 2020* have recently reported that the induction of DNA breaks by two clastogenic G4 ligands, CX-5461 and pyridostatin (PDS), may result from a TOP2-poisoning-like mechanism, forming TOP2cc, but the respective contribution of TOP2A/TOP2B to the clastogenic activity of both G4 ligands is still unclear. Moreover, a recent report argues that cytotoxicity of CX-5461 may rely on irreversible inhibition of RNA-Pol I transcription initiation (*Mars et al., 2020*). Here, we report a differential contribution of TOP2A and TOP2B to DNA break production in response to G4 ligands PDS and CX-5461 and identify TOP2A as a major effector of cytotoxicity induced by the two G4 ligands. We also report that transcription plays an essential role in the production of TOP2-dependent DNA breaks induced by both G4 ligands. We show that G4 ligand-induced DSBs are countered by TOP1 that limits co-transcriptional G4 formation and by factors that promote transcriptional elongation. Taken together, our results support the concept that CX-5461 and PDS act as DNA structure-driven TOP2 poisons at G-rich transcribed genomic regions.

## Results

### *TOP2A* mutations confer resistance to clastogenic G4 ligands CX-5461 and pyridostatin

The chemical compound CX-5461, currently in phase I/II clinical trials for cancer treatments (*Khot et al., 2019*), was first described as an RNA-Pol I inhibitor (*Haddach et al., 2012*). Although the cytotoxic effects induced by this compound have been related to rDNA-transcription inhibition (*Mars et al., 2020*; *Negi and Brown, 2015*), CX-5461 has also been shown to be a potent G4 stabilizer and to provoke rapid induction of DSBs through a TOP2-dependent mechanism (*Bruno et al., 2020*; *Olivieri et al., 2020*; *Xu et al., 2017*). In order to more clearly define how CX-5461 mediates its cytotoxicity, we adopted an unbiased approach based on the selection and characterization of cells resistant to this drug. To do this, human near haploid HAP1 cells were randomly mutagenized with ethyl methane sulfonate (EMS) based on previous work (*Forment et al., 2017*) and clones resistant to a lethal CX-5461 concentration of 0.3 μM were isolated (CX-5461-resistant [CXR] clones). Resistance of seven of these clones to CX-5461 was further confirmed by cell survival assays showing $IC_{50}$ values on CXR clones ranging from 0.22 to 0.34 μM CX-5461, corresponding to an average ninefold increase in the $IC_{50}$ value compared to the 0.03 μM $IC_{50}$ on wild-type HAP1 cells (WT) (see *Figure 1A* and *Supplementary file 1*). In addition, CXR clones did not show cross-resistance to the unrelated drug and efflux-pumps substrate nocodazole (*Kasap et al., 2014*), excluding a multidrug resistance (MDR) phenotype (*Supplementary file 1*).

Inspired by previous work (*Wacker et al., 2012*), we analyzed the selected CXR clones through a global RNA-sequencing approach (RNA-seq) to identify non- and mis-sense mutations in coding genes that could account for the observed resistance. Around eight genes per clone were found with non- or mis-sense mutations through this approach (*Figure 1B* and *Supplementary file 2*). Unbiased analysis of genes mutated in several clones revealed that each clone, except CXR #A2 and CXR #A6, carried a homozygous mutation in the *TOP2A* gene, encoding for the TOP2A protein (*Figure 1B*). Manual analysis of the sequencing data for the *TOP2A* gene confirmed these mutations and revealed that the CXR #A2 clone carried the S654N mutation, while the CXR #A6 clone carried a homozygous mutation of the first nucleotide of the last intron, resulting in intron retention and replacement of the last 42 TOP2A amino acids, carrying the nuclear localization signal (NLS), by 18 unrelated amino acids (*Figure 1—figure supplement 1A–C*). From these analyses, *TOP2A* emerged as the only gene with coding mutations in all resistant clones. Four clones had a mutation in the ATPase domain (F85I), while two had mutations in the DNA-binding region (S654N and L703I) (*Figure 1C*). Immunoblotting analysis revealed that none of the identified TOP2A mutations resulted in the loss of TOP2A protein, in agreement with its essential function in proliferating cells (*Akimitsu et al., 2003*; *Carpenter and Porter, 2004*). In addition, TOP2B expression level was unaffected in these clones (*Figure 1D*).

To test whether the catalytic activity of the TOP2A protein was altered in CXR clones, we determined the sensitivity of WT and CXR HAP1 clones to the TOP2 poison etoposide (ETP), a chemotherapeutic drug that acts by stabilizing TOP2cc and the cytotoxicity of which is therefore dependent on TOP2 activity (*Walker and Nitiss, 2002*). All CXR presented a strong resistance to ETP with resistance indexes ranging from 8- to 19-fold relative to control cells (*Figure 2C* and *Supplementary file 1*). These results support the idea that *TOP2A* point mutations in CXR clones reduce TOP2A activity. To confirm this, we adapted a heparin-based extraction protocol (*de Campos-Nebel et al., 2016*) coupled with immunoblotting to monitor the accumulation of TOP2Acc following ETP treatment. In this assay, TOP2 molecules not covalently attached to DNA are extracted by heparin to a soluble fraction, while TOP2ccs are resistant to heparin and can be analyzed by immunoblotting of the insoluble pellet fraction. As shown in *Figure 2—figure supplement 1*, the point mutations present in clones CXR #A6 and CXR #A1 decrease the amount of TOP2Acc following ETP treatment.

In parallel, using the same genetic approach, we isolated F14R HAP1 clones resistant to a lethal concentration of 30 nM F14512, a potent and selective TOP2A poison (*Bombarde et al., 2017*). Targeted sequencing of TOP2A cDNA in five F14R clones revealed five different *TOP2A* mutations, confirming that TOP2A is the main mediator of F14512 cytotoxicity (*Bombarde et al., 2017*). One mutation was found in the transducer domain (W414L), while the four others lay in the DNA-binding domain (Y590H, P593S, G777D, and P890L; *Figure 1C*). All of these mutations were different from

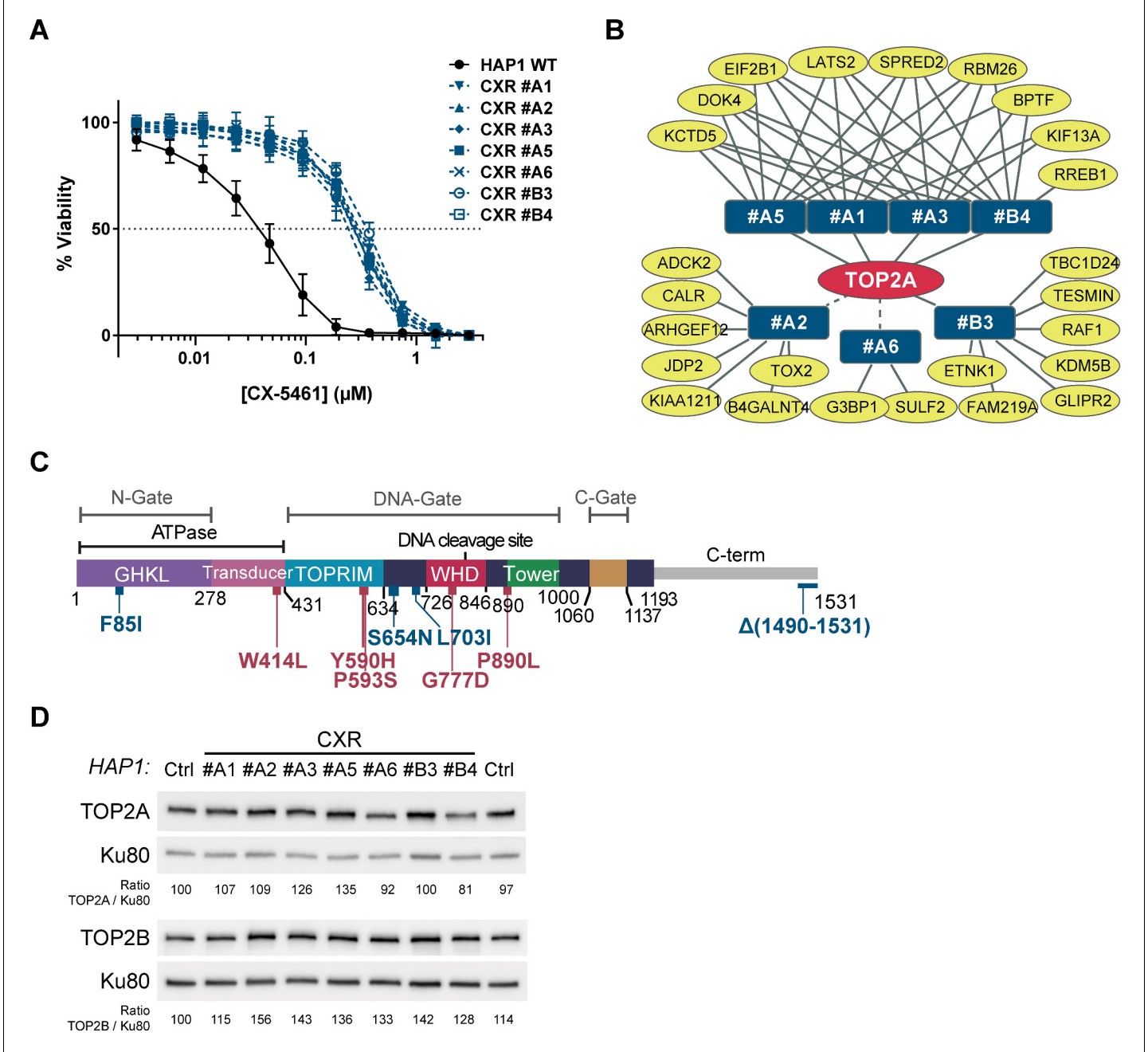

**Figure 1.** Role for topoisomerase 2α (TOP2A) in the cell toxicity of CX-5461. (A) Viability assay on wild-type (WT) and seven CX-5461-resistant (CXR) HAP1 clones treated with CX-5461. Error bars represent SD from the means, $n \geq 3$ independent experiments. (B) Representation of genes with non- and mis-sense mutations identified in CXR clones. Mutated genes identified in resistant clones (blue rectangles) are represented as yellow ovals. The presence of a *TOP2A* mutation in all resistant clones is schematized by the central position of TOP2A gene in the red oval. Solid and dashed lines represent respectively mutations characterized through an unbiased or manual analysis of RNA-seq data. (C) Linear schematic of TOP2A domains. Each domain is labeled and described by bordering residue numbers. TOP2A mutations present in CX-5461 or F14512-resistant clones are indicated in blue or red, respectively. (D) Immunoblotting analysis of whole-cell extracts from WT (Ctrl) and CXR HAP1 cells. Relative protein levels of TOP2A and TOP2B were quantified, normalized to KU80 level, and set to 100 in Ctrl cells.

The online version of this article includes the following source data and figure supplement(s) for figure 1:

**Source data 1.** Raw unedited image and uncropped figure of the blot of the western blot from *Figure 1*.

**Figure supplement 1.** Molecular characterization of topoisomerase 2α (TOP2A) mutations in CXR cells.

**Figure supplement 1—source data 1.** Raw unedited image and uncropped figure of the blot of the western blot from *Figure 1—figure supplement 1*.

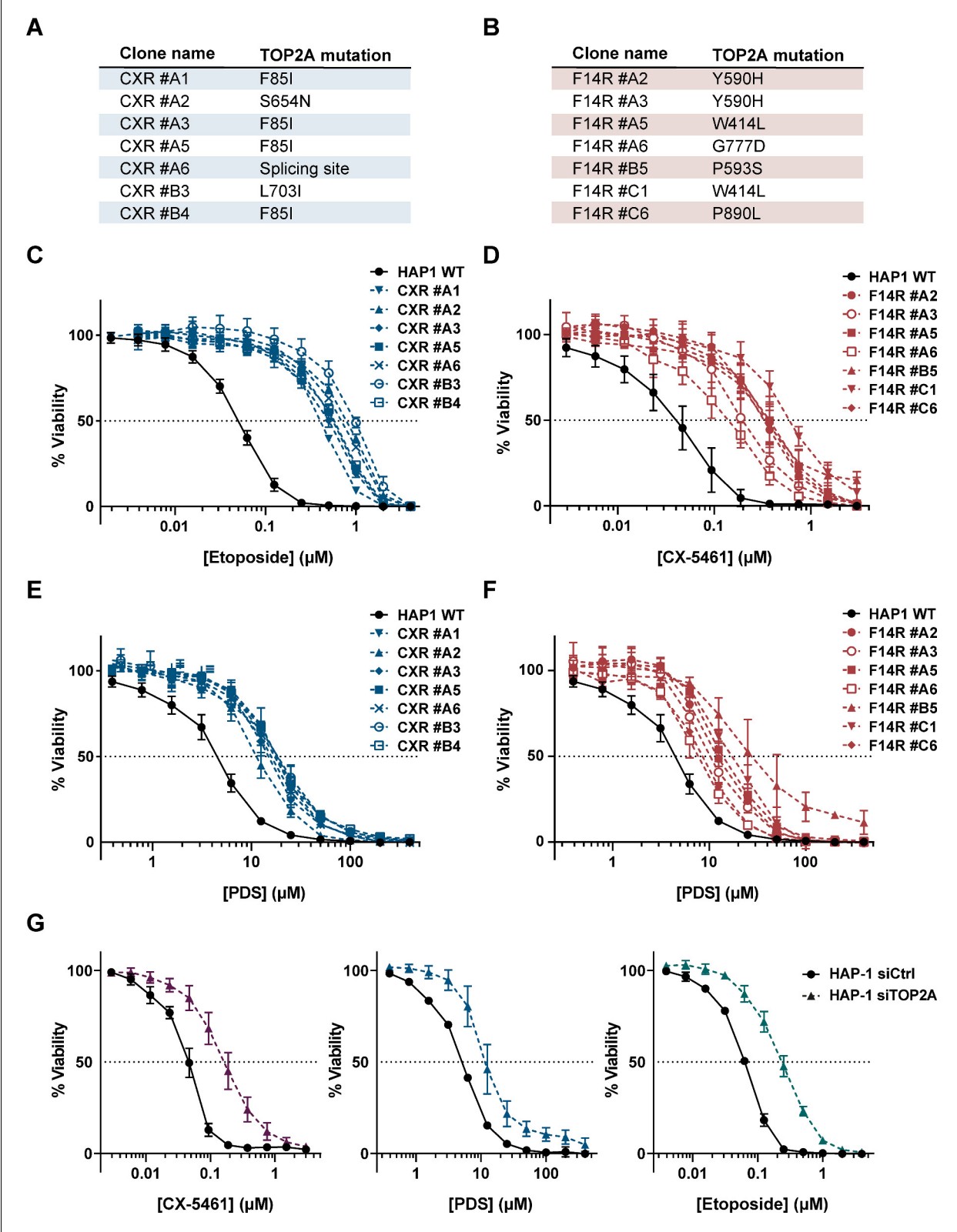

**Figure 2.** Impact of topoisomerase 2α (TOP2A) in the cell toxicity of topoisomerase poisons and G-quadruplex (G4) ligands. (**A**, **B**) Summary of mutations found in CX-5461-resistant (CXR) (**A**) and F14R (**B**) clones. (**C**, **E**) Viability assay of CXR cells treated with etoposide (ETP) (**C**) and the G4 ligand pyridostatin (PDS) (**E**). (**D**, **F**) Viability assay of F14512-resistant cells (F14R) treated with the G4 ligands CX-5461 (**D**) and PDS (**F**). (**G**) Viability assay of

*Figure 2 continued on next page*

*Figure 2 continued*

TOP2A knock-down HAP1 cells treated with G4 ligand CX-5461 (left panel), PDS (central panel), and ETP (right panel). Error bars represent SD from the means, $n \geq 3$ independent experiments.

The online version of this article includes the following source data and figure supplement(s) for figure 2:

**Source data 1.** Raw unedited image and uncropped figure of the blot of the western blot from *Figure 2*.
**Figure supplement 1.** Analysis of etoposide-induced topoisomerase 2α (TOP2A) cleavage complexes in wild-type and CX-5461-resistant (CXR) HAP1.
**Figure supplement 1—source data 1.** Raw unedited image and uncropped figure of the blot of the western blot from *Figure 2—figure supplement 1*.
**Figure supplement 2.** Cell proliferation and DNA repair studies in wild-type (WT) and CX-5461-resistant (CXR) HAP1 cells.
**Figure supplement 2—source data 1.** Raw unedited image and uncropped figure of the blot of the western blot from *Figure 2—figure supplement 2*.

the ones found in the CXR clones, which suggests that, despite both acting through TOP2A, CX-54161 and F14512 affect TOP2A activity differently Strikingly, cell survival assays clearly demonstrated that all of the F14R cells were cross-resistant to CX-5461 (*Figure 2D*), while showing no resistance to nocodazole (*Supplementary file 1*), with $IC_{50}$ values for CX-5461 3.6- to 16.8-fold higher than the $IC_{50}$ value of CX-5461 in WT cells (*Figure 2D* and *Supplementary file 1*), supporting the concept that the *TOP2A* mutations found in F14R clones also conferred resistance to the G4 ligand CX-5461.

Importantly, population doubling time determination of five clones carrying different *TOP2A* mutations and flow cytometry analyses of 5-ethynyl-2′-deoxyuridine (EdU) incorporation in CXR #A6 and F14R #C1 cells revealed no significant differences in the proliferation rate of *TOP2A* mutated clones as compared to WT cells (*Figure 2—figure supplement 2A, B*).

In addition, the survival of WT and *TOP2A* mutated clones to camptothecin, a DNA TOP1 poison, or calicheamicin, a radio-mimetic compound, was not significantly different. Since camptothecin induces toxic replication-associated DSBs, which are mainly repaired by HR, while calicheamicin induces DSB in all cell cycle phases, which are repaired by both NHEJ and HR (*Elmroth et al., 2003*; *Furuta et al., 2003*), these results indicate that DSB repair mechanisms are proficient in *TOP2A* mutated cells (*Figure 2—figure supplement 2C*). Supporting this conclusion, the sensitization to calicheamicin induced by an inhibitor of the NHEJ factor DNA-PK was similar in WT and TOP2A mutated cells (*Figure 2—figure supplement 2D*) and the kinetics of DSB repair after X-ray irradiation measured by flow cytometry analysis of the γH2AX signal were identical between WT and mutant cells (*Figure 2—figure supplement 2E*). Finally, western blot analysis showed that TOP1 levels were unaffected in *TOP2A* mutated cells (*Figure 2—figure supplement 2F*). Altogether, these results indicate that the resistance to CX-5461 observed in CXR and F14R cells is not associated with major changes in proliferation or DNA repair mechanisms.

To investigate whether CX-5461 resistance induced by TOP2A mutations extends to other G4 stabilizers, we tested CXR mutants for their cross-resistance to pyridostatin (PDS), one of the best characterized G4 ligands so far (*Rodriguez et al., 2008*). In cells, PDS treatment, similar to CX-5461, induces a rapid accumulation of DSBs, but in contrast to CX-5461, PDS does not affect RNA-Pol I activity (*Rodriguez et al., 2012*). Cell survival assays showed that all CXR clones were cross-resistant to PDS ($IC_{50}$ values for PDS 2.7- to 4.4-fold higher than the $IC_{50}$ value of PDS on WT cells) but with lower resistance indexes than those observed for CX-5461 and F14512 (*Figure 2E* and *Supplementary file 1*). More remarkably, cross-resistance studies established that all F14R clones were also resistant to PDS (*Figure 2F* and *Supplementary file 1*). The role of TOP2A in the cytotoxicity of CX-5461 and PDS was further supported by the marked resistance to both CX-5461, PDS, and ETP (as a control) conferred by small-interfering RNA-mediated depletion of TOP2A in HAP1 cells, with $IC_{50}$ values similar to those observed for mutant cells (*Figure 2G*).

In cells, stabilization of G4 by ligands has been shown to provoke cell growth modifications relying on DNA and RNA transaction alterations (*Varshney et al., 2020*). The acridine derivative RHPS4, a potent G4 ligand, affects both telomere maintenance and mitochondrial DNA replication (*Falabella et al., 2019*; *Salvati et al., 2007*), while two selective bisquinolinium G4 ligands, 360A and PhenDC3, impact on telomeres, gene expression, and RNA metabolism (*Dumas et al., 2021*; *Halder et al., 2012*; *Pennarun et al., 2005*). Through performing cell viability with RPHPS4 and PhenDC3, we revealed that the TOP2A mutant cells do not have significantly different modifications

in response to these two chemically unrelated ligands as compared to WT cells (*Figure 2—figure supplement 2G*).

Altogether, these results indicate that TOP2A is specifically involved in the cytotoxic effect induced by the two G4 stabilizers CX-5461 and PDS.

## Rapid accumulation of DNA DSBs upon G4 ligand treatment depends on TOP2 activity

In human cells, short treatments with PDS or CX-5461 induce rapid production of the DSB markers γH2AX and 53BP1 foci (*Rodriguez et al., 2012*; *Xu et al., 2017*; *Figure 3A*, *Figure 3—figure supplement 1A*, and *Figure 3—figure supplement 2A*). G4-dependent γH2AX foci production is increased in cells incubated with the DNA-PK inhibitor NU7441 (DNA-PKi; *Figure 3—figure supplement 2B*), indicating that a substantial number of G4-induced DNA breaks are repaired through the DNA-PK-dependent NHEJ pathway, the major DSB repair pathway in human cells (*Pannunzio et al., 2018*). Considering the main contribution of DSBs to the cytotoxic effect of several anticancer agents and having shown that TOP2A activity determines the cytotoxic effect of G4 ligands (*Figures 1* and *2*), we evaluated the role of TOP2A in DSB production upon PDS and CX-5461 treatments. First, in CXR cells carrying different mutations in the TOP2A protein, clones CXR #A1 (F85I), #A2 (S654N), and #A6 (intron retention), γH2AX production was significantly reduced as compared to control cells (*Figure 3A*). We assessed the role of TOP2 activities on DSBs production induced by PDS and CX-5461 treatments in HeLa cells by studying the impact of the TOP2 catalytic inhibitor BNS-22 (*Kawatani et al., 2011*; *Figure 3B*, *Figure 3—figure supplement 1B*, and *Figure 3—figure supplement 2C*). Pre-incubation with BNS-22 significantly decreased the number of γH2AX foci in cells treated with both G4 ligands, thereby demonstrating the major role of TOP2 catalytic activity in the production of DNA breaks following G4 ligand treatments. To evaluate the respective contribution of TOP2A and TOPB2B proteins in the formation of G4-dependent DSBs, we analyzed γH2AX production in HeLa cells transfected by siRNA against each TOP2 as compared to cells transfected with control siRNA. Immunofluorescence studies showed that in TOP2A-depleted cells γH2AX production was abolished upon PDS treatment and strongly reduced upon CX-5461 treatment (*Figure 3C*, *Figure 3—figure supplement 1C*, and *Figure 3—figure supplement 2D*). In contrast, TOP2B depletion did not impact γH2AX production upon PDS treatment, while it reduced γH2AX production upon CX-5461 treatment to a similar extent to that induced by TOP2A depletion. Moreover, simultaneous siRNA-mediated knock-down of TOP2A and TOP2B further decreased γH2AX production by CX-5461 when compared to the effect of separate siRNAs (*Figure 3C* and *Figure 3—figure supplement 1C*). Altogether, these results indicate that TOP2A activity is the main effector of DSBs by clastogenic G4 ligands in human cells, with TOP2B variably contributing depending on the ligand class.

## TOP2-dependent DSBs induced by CX-5461 and PDS are associated with G4 structures in cells

To investigate whether TOP2-dependent DSBs induced by CX-5461 and PDS are associated with G4 structures, we adapted the heparin pre-extraction protocol to immunofluorescence in order to visualize the formation of covalent TOP2A-DNA complex (TOP2Acc) in cells. To validate this novel assay, TOP2A and γH2AX signals were quantified upon ETP treatment. As shown in *Figure 4A, B*, ETP provoked a strong increase in TOP2A signal that correlated with a similar increase in γH2AX foci (*Figure 4A, B* and *Figure 4—figure supplement 1A, B*). Using the same approach, we observed a significant increase of TOP2Acc and γH2AX signals in cells treated with CX-5461 and PDS (*Figure 4A, B* and *Figure 4—figure supplement 1A, B*), supporting the idea that these G4 ligands mediate their cytotoxic effect through stabilizing TOP2Acc.

The bisquinolinium compound 360A is a potent and selective G4 binder that exhibits a lack of clastogenic activity in most human cells as compared to CX-5461 and PDS (*Figure 4A* and *Figure 4—figure supplement 1A*). Assuming that the observed formation of TOP2Acc upon CX-5461 and PDS treatment results from the stabilization of G4 structures, we hypothesized that the binding of 360A to the same structures would impair that of CX-5461 and PDS and thereby inhibit the formation of TOP2Acc in cells treated with the latter clastogenic ligands. As shown in *Figure 4B* (and *Figure 4—figure supplement 1B*), 360A pre-treatment strongly reduced the induction of TOP2Acc by both

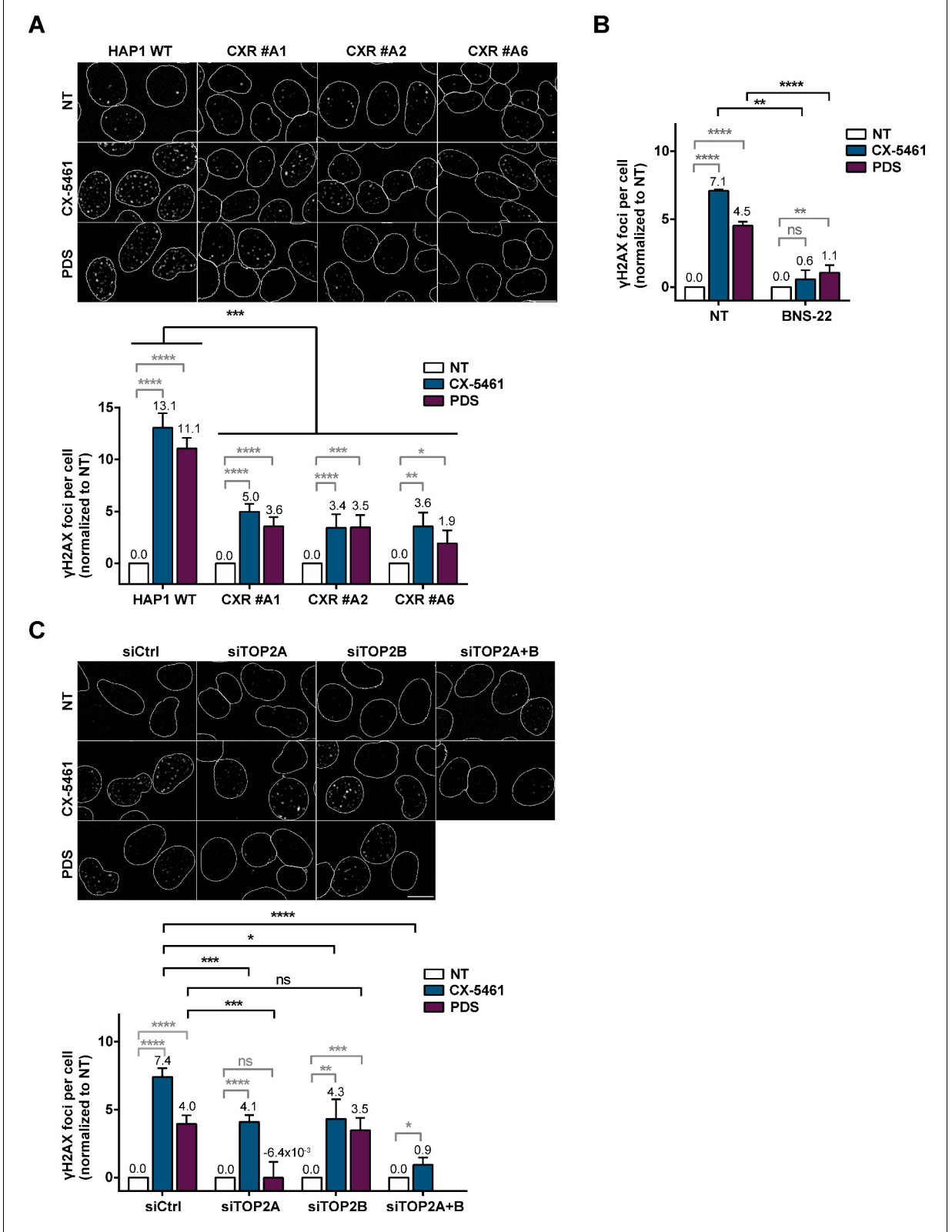

**Figure 3.** Role of topoisomerase 2 (TOP2) proteins in DNA breaks production by G-quadruplex (G4) ligands CX-5461 and pyridostatin (PDS). (**A**) Representative images (upper panel) and quantification (bottom panel) of γH2AX foci detected in HAP1 wild-type (WT) and CX-5461-resistant (CXR) cells. (**B**) Quantification of γH2AX foci detected after PDS or CX-5461 treatment in HeLa cells pre-treated with the TOP2 catalytic inhibitor BNS-22. (**C**) Representative images (upper panel) and quantification (bottom panel) of γH2AX foci detected in HeLa cells transfected with control (Ctrl),

*Figure 3 continued on next page*

*Figure 3 continued*

topoisomerase 2α (TOP2A), and/or topoisomerase 2β (TOP2B) siRNAs and treated with PDS or CX-5461. For all the experiments, cells were incubated with 0.2 μM CX-5461 or 20 μM PDS for 4 hr. For experiments with BNS-22, a 5 μM pre-treatment was performed for 30 min prior to addition of PDS. Quantification of γH2AX foci per cell was performed on n > 165, n > 101, and n > 105 nuclei for each condition, respectively, in (A), (B), and (C). Error bars represent SD from the means, n ≥ 3 independent experiments. p values were calculated using an unpaired multiple Student's *t* test. ns: p>0.05; *p<0.05; **p<0.01; ***p<0.001; ****p<0.0001.

The online version of this article includes the following source data and figure supplement(s) for figure 3:

**Figure supplement 1.** Not normalized data from *Figure 3*. Role of topoisomerase 2 (TOP2) proteins in DNA breaks production by G-quadruplex (G4) ligands CX-5461 and pyridostatin (PDS).

**Figure supplement 1—source data 1.** Raw unedited image and uncropped figure of the blot of the western blot from *Figure 3—figure supplement 1*.

**Figure supplement 2.** Role of topoisomerase 2 (TOP2) proteins in DNA breaks production by G-quadruplex (G4) ligands CX-5461 and pyridostatin (PDS).

clastogenic G4 ligands, without impacting on the TOP2Acc signal induced by ETP. As expected, inhibition by 360A pre-treatment of TOP2Acc formation resulted in a significant reduction in the γHA2X signal induced by clastogenic G4 ligands, without impacting on the formation of DSBs in ETP-treated cells (*Figure 4A* and *Figure 4—figure supplement 1A*).

To correlate TOP2Acc with stabilization of G4 structures in cells treated by CX-5461 and PDS, we used high-resolution imaging (3D-SIM) to perform colocalization studies of TOP2Acc with G4 as visualized with the selective anti-G4 antibody, BG4 (*Biffi et al., 2013*). As shown in *Figure 4C* (and *Figure 4—figure supplement 1C* ), CX-5461 and PDS treatments induced a significant increase in the percentage of BG4 signals colocalizing (*Figure 4D*) with TOP2Acc and induced a significant increase in the percentage of TOP2Acc signals colocalizing with G4 structures. Furthermore, analysis of BG4 signals showed a strong increase of G4 structures in all the treated conditions (*Figure 4—figure supplement 2*). It is noteworthy that the increase in TOP2Acc-G4 colocalizations was not observed in ETP-treated cells, showing that the percentage of colocalization does not result from a global increase of TOP2Acc or BG4 signals induced by the different treatments (*Figure 4—figure supplement 2*). The increased BG4 signal observed in ETP-treated cells probably results from a global increase of DNA supercoiling by ETP, which would promote G4 formation as shown in several other studies (*Sekibo and Fox, 2017*; *Sun, 2010*; *Sun and Hurley, 2009*).

Altogether, our data support the conclusion that TOP2-dependent DSBs induced by CX-5461 and PDS are associated with G4 structures in cells.

## TOP2-dependent DSBs induced by G4 ligands are transcription-dependent

We observed that PDS-induced DSB markers appeared in all cell cycle phases, indicating that PDS-induced DNA damage is not strictly S-phase dependent (*Figure 5—figure supplement 1A*; *Rodriguez et al., 2012*). Using the DNA-base analog EdU to visualize DNA synthesis, we confirmed that DNA damage induction by PDS and CX-5461 also occurs outside of S phase (*Figure 5A* and *Figure 5—figure supplement 2A*; *Rodriguez et al., 2012*). Furthermore, comparison of the number of G4 ligand-induced γH2AX foci in EdU-negative cells and in the total cell population indicated that DNA replication-independent processes were very efficient for the production of DSBs by PDS (*Figure 5A* and *Figure 5—figure supplement 2A*). Accordingly, DSB production by PDS in HeLa cells was strongly reduced upon inhibition of RNA-Pol II-dependent transcription by 5,6-dichloro-1-b-D-ribofuranosylbenzimidazole (DRB), an inhibitor of critical phosphorylations in the RNA-Pol-II C-terminal domain (*Yankulov et al., 1995*; *Figure 5B* and *Figure 5—figure supplement 2B*). Importantly, immunofluorescence analysis with the G4-specific antibody BG4 showed that inhibition of RNA-Pol II-dependent transcription did not impede the accumulation of G4 structures caused by PDS treatment (*Figure 5—figure supplement 1B*).

DRB also reduced DSB production upon CX-5461 treatment, albeit to a lesser extent than with PDS (*Figure 5B* and *Figure 5—figure supplement 2B*). Thus, we next investigated the contribution of RNA-Pol I-dependent transcription to γH2AX production upon CX-5461 treatment. BHM21, a DNA intercalator showing a preferential binding to GC-rich sequences (*Peltonen et al., 2014*) but not to G4 structures (*Xu et al., 2017*), is a potent RNA-Pol I inhibitor that causes the proteasome-

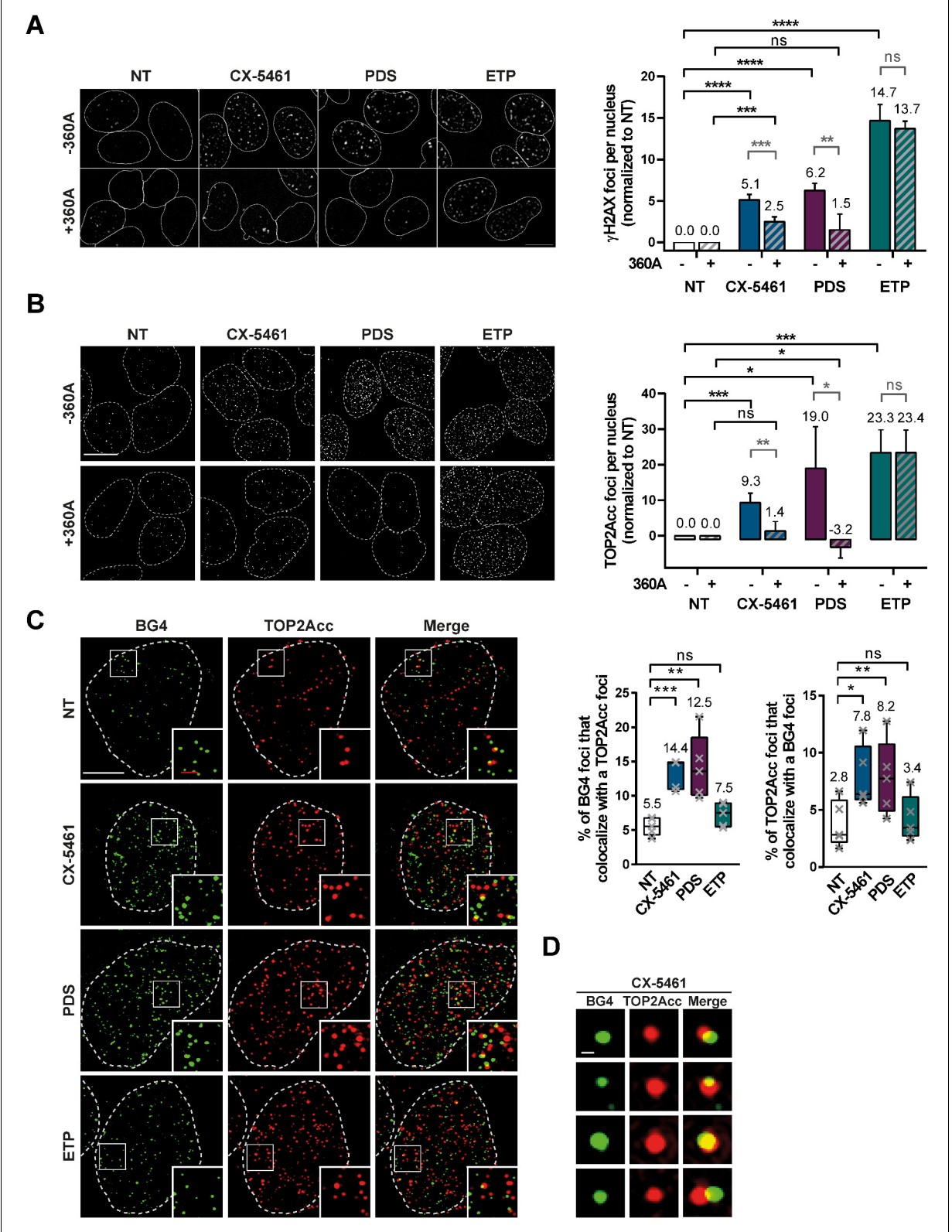

**Figure 4.** TOP2Acc induced by CX-5461 and pyridostatin (PDS) are associated with G-quadruplex (G4) structures in cells. Representative images (left panel) and quantification (right panel) of γH2AX (**A**) and TOP2Acc (**B**) foci detected in HeLa cells treated for 4 hr with G4 ligands CX-5461 (0.2 μM) and PDS (20 μM) or the topoisomerase 2-poison etoposide (ETP) (3.75 μM). For the conditions with 360A compound, a 20 μM treatment was performed for 3 hr prior to PDS, CX-5461, and ETP treatment and renewed for the duration of the treatment. Quantification of γH2AX foci and TOP2Acc foci per cell

*Figure 4 continued on next page*

Figure 4 continued

in (A) and (B) was performed on $n \geq 219$ and $n \geq 144$, respectively, for each condition. Error bars represent SD from the means of $n \geq 3$ independent experiments. Bar: 10 μm. (C) Colocalization of TOP2Acc and BG4 fluorescence signals in HeLa cells treated for 15 min with G4 ligands CX-5461 (0.2 μM) and PDS (20 μM) or the topoisomerase 2-poison ETP (3.75 μM). Representative pictures on the left panel correspond to maximum intensity projections of 20 3D-SIM Z-stacks (interval 0.091 μm) Quantifications are shown on the right panel. Error bars represent SD from the means of $n = 5$ independent experiments in which $n \geq 8$ nucleus were quantified for each condition. Bars: white, 5 μm; red, 1 μm. (D) Representative pictures of BG4 foci that co-localize with TOP2Acc foci. Bar: 200 nm. p values in (A–C) were calculated using an unpaired multiple Student's t test.

The online version of this article includes the following figure supplement(s) for figure 4:

**Figure supplement 1.** Not normalized data from *Figure 4*.
**Figure supplement 2.** BG4 and TOP2Acc signals quantification in HeLa cells.

dependent degradation of RPA194, the large catalytic subunit protein of RNA-Pol I holoenzyme (*Peltonen et al., 2014*). In HeLa cells, pre-treatment with BMH21 significantly reduced the formation of γH2AX signals induced by CX-5461 treatment (*Figure 5C* and *Figure 5—figure supplement 2C*). Remarkably, the concomitant inhibition of RNA-Pol II and RNA-Pol I-dependent transcription strongly reduced DSB production by this compound (*Figure 5D* and *Figure 5—figure supplement 2D*).

Altogether, these data support a key role of transcription in the production of DSBs following the stabilization of G4 structures by PDS and CX-5461.

In human cells, CX-5461 and PDS have been shown to provoke a rapid accumulation of R-loops (*De Magis et al., 2019*; *Sanij et al., 2020*; *Shen et al., 2017*; *Duquette et al., 2004*), which have been associated with the production of transcription-dependent DSBs and genomic instability (*De Magis et al., 2019*). Thus, in order to investigate the impact of R-loops in the DNA damage production by PDS and CX-5461, we used a U2OS cell line expressing the *Escherichia coli* RNaseHI, which cleaves and removes R-loops, under the control of a doxycycline-inducible promoter (*Britton et al., 2014*). Upon RNaseHI expression, we observed a significant decrease of DNA damage signals induced by PDS and CX-5461 (*Figure 5E* and *Figure 5—figure supplement 2E*), indicating that the production of R-loop structures contributes to the formation of DNA breaks following G4 stabilization. Altogether, these results indicate that transcription elongation plays a key role in the production of DNA damage by both G4 ligands.

## TOP2-dependent DSBs induced by G4 stabilizers are countered by TOP1 and factors promoting transcription elongation

Since the formation of G4 structures is linked to negative supercoiling, RNA-Pol II pausing, and R-loop formation, three transcription-dependent processes involving TOP1 activity (*Kim and Jinks-Robertson, 2017*), we evaluated the role of TOP1 enzyme in the formation of DSBs following treatment with G4 ligands. Strikingly, shRNA-mediated depletion of TOP1 protein in human cells caused a significant increase in DNA damage induced by CX-5461 and PDS (*Figure 6A* and *Figure 6—figure supplement 1A*). Furthermore, immunofluorescence studies showed that the increase in DNA damage signals in TOP1-depleted cells was dependent on TOP2 activity since BNS22 pre-treatment in these cells blocked the induction of γH2AX signals by CX-5461 and PDS (*Figure 6—figure supplement 2A*). Confirming the role of TOP2, DNA break formation upon PDS treatment in TOP1 knock-down cells was also impaired by TOP2A depletion (*Figure 6—figure supplement 2B*).

EdU staining revealed that enhanced DNA damage production in TOP1 knock-down cells is not restricted to S phase of the cell cycle (*Figure 6—figure supplement 3B*). Moreover, DRB pre-treatment abrogated PDS-induced DNA damage in TOP1-depleted cells, indicating that they were fully dependent on transcription (*Figure 6—figure supplement 3A*). Finally, the depletion of TOP1 in HeLa cells caused a significant increase in the cytotoxic effect of both PDS and CX-5461 that was reverted by pre-treatment with DRB or DRB plus BMH21 for PDS and CX-5161, respectively (*Figure 6B* and *Figure 6—figure supplement 1B*). Altogether, these results support the hypothesis that a key factor in the mechanism of TOP2-mediated DNA damage produced by clastogenic G4 ligands is the accumulation of topological stresses provoked by transcription that are countered by TOP1. Consistent with these findings, immunofluorescence analysis showed that TOP1 knock-down provoked a significant increase of the BG4 signal in human cells (*Figure 6C*, *Figure 6—figure supplement 1C*, and *Figure 6—figure supplement 3C*) with no cumulative effect of PDS and TOP1

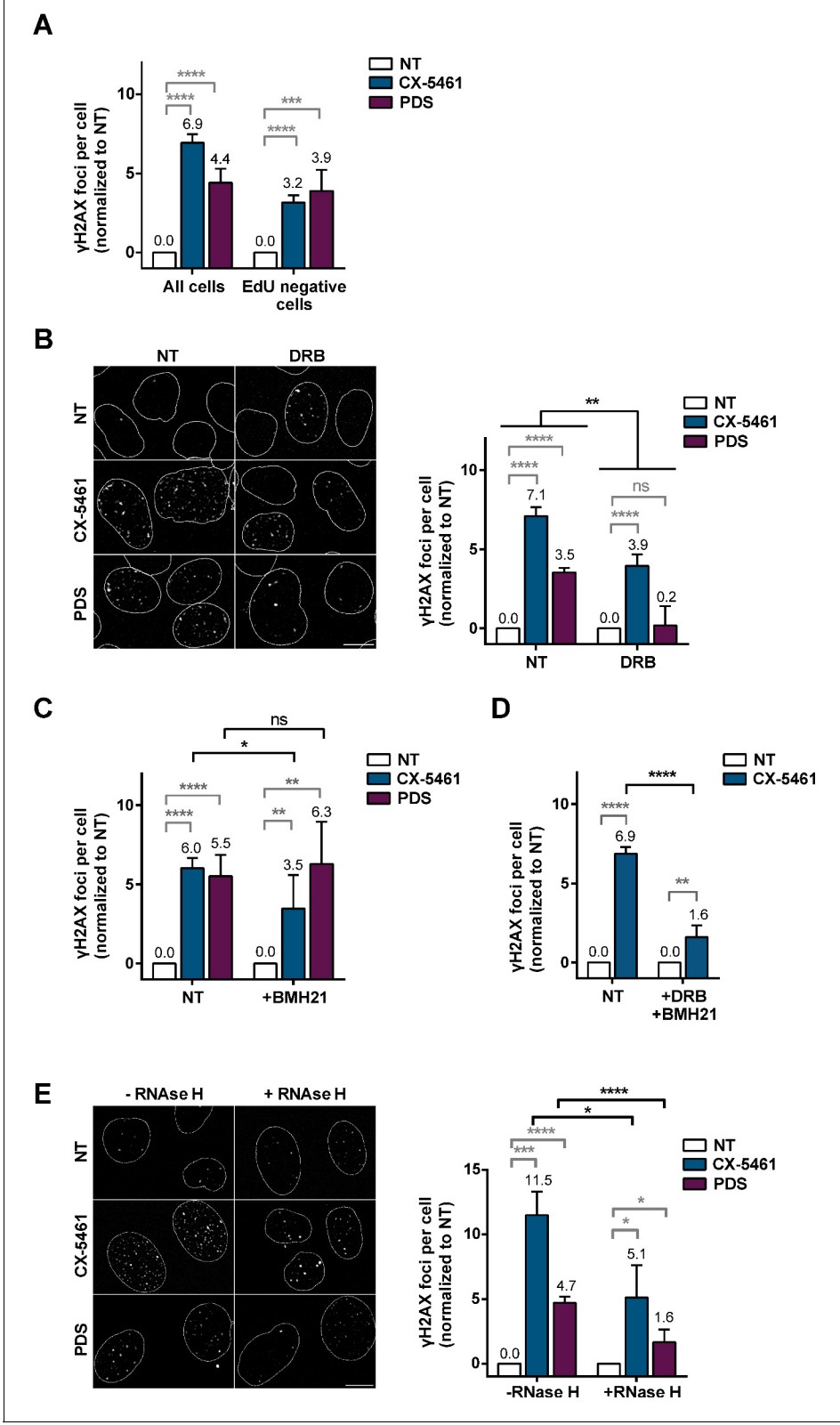

**Figure 5.** Role of RNA-Pol II-dependent transcription in DNA breaks production by G-quadruplex (G4) ligands CX-5461 and pyridostatin (PDS). Quantification of γH2AX foci in HeLa cells treated with 0.2 μM CX-5461 or 20 μM PDS for 4 hr in the presence of 5-ethynyl-2'-deoxyuridine (EdU). (A). Representative images (left panel) and quantification (right panel) of γH2AX foci fluorescence signal detected in HeLa cells pre-treated with the RNA-Pol II inhibitor 5,6-dichloro-1-b-D-ribofuranosylbenzimidazole (DRB) (B), RNA-Pol I inhibitor BMH21 (C) or DRB plus BMH21 (D) prior to addition of 0.2

*Figure 5 continued on next page*

*Figure 5 continued*

μM CX-5461 or 20 μM PDS for 4 hr. (**E**) Representative images (left panel) and quantification (right panel) of γH2AX foci upon PDS or CX-5461 treatment (8 hr) in RNaseH1-mCherry U2OS-expressing cells. DRB or BMH21 were added 1 hr before CX-5461 or PDS treatment. RNaseH1-mCherry expression in U2OS cells was induced 14 hr prior to PDS treatment. γH2AX foci per cell was performed on $n > 165$ nuclei for each condition in (**A**) and $n > 42$ nuclei for each condition in (**B**). Error bars represent SD from the means, $n \geq 3$ independent experiments. p values were calculated using an unpaired multiple Student's *t* test. ns: $p > 0.05$; *$p < 0.05$; **$p < 0.01$; ***$p < 0.001$; ****$p < 0.0001$.

The online version of this article includes the following figure supplement(s) for figure 5:

**Figure supplement 1.** Role of RNA-Pol II-dependent transcription in DNA breaks production by G-quadruplex (G4) ligands CX-5461 and pyridostatin (PDS).

**Figure supplement 2.** Not normalized data from *Figure 5*. Role of RNA-Pol II-dependent transcription in DNA breaks production by G-quadruplex (G4) ligands CX-5461 and pyridostatin (PDS).

depletion on BG4 signal. We concluded that the accumulation of transcription-dependent negative supercoiling resulting from TOP1 depletion has a major impact on G4 formation. In line with this result, kinetic studies of γH2AX production in PDS-treated cells showed that TOP1 depletion significantly accelerates the formation of PDS-induced DNA damage signals compared to control cells (*Figure 6D* and *Figure 6—figure supplement 1D*), indicating that TOP1 depletion facilitates DSB production by G4 ligands.

During transcription, TOP1 activity is enhanced by the BRD4-dependent phosphorylation of RNA-Pol II CTD (*Baranello et al., 2016*). BRD4 also associates with the positive transcription elongation factor (PTEF) to promote transcription and release RNA-Pol II from paused sites (*Jang et al., 2005*; *Yang et al., 2005*). In cells, BRD4 depletion provokes R-loop accumulation that drives transcription-dependent DNA damage (*Kim et al., 2019*; *Lam et al., 2020*). Interestingly, we showed through immunofluorescence analysis that the inhibition of BRD4 activity by the potent inhibitor JQ1 provoked a significant increase of DNA breaks induced by PDS (*Figure 6E* and *Figure 6—figure supplement 1E*), indicating that inhibition of transcription elongation facilitates the formation of DSBs upon G4 ligand treatment. Altogether, our results strongly suggest that TOP1 antagonizes G4 ligand action through a transcription-dependent mechanism that is related to RNA polymerase stalling.

## Discussion

In this study, using an unbiased genetic approach, we identified the TOP2A protein as the main effector of the cytotoxicity of two clastogenic G4 ligands: CX-5461 and PDS. Our study highlights the strength of the genetic approach we applied here, relying on chemical mutagenesis in a haploid background (*Forment et al., 2017*). Indeed, despite TOP2A being essential in proliferating cells (*Akimitsu et al., 2003*; *Carpenter and Porter, 2004*), we were able to readily isolate CX-5461-resistant clones carrying *TOP2A* mutations allowing direct identification of its crucial role in DSB induction upon G4 stabilization. This would not have been possible through loss of function screens, for example, using CRISPR/Cas9 or insertional mutagenesis. In addition, we produced novel mutations of *TOP2A* conferring resistance to both CX-5461 and F14512 that could be useful to obtain insights into TOP2A biology. It is noteworthy that these nine-point mutations, separating TOP2A essential function from its role in G4-induced DSB production, are broadly distributed throughout the *TOP2A* coding sequence and functional domains. This indicates that, as observed for TOP2 poisons in a yeast complementation system (*Blower et al., 2019*), resistance to CX-5461 can be obtained through different protein modifications and is not strictly dependent on the catalytic function. This is especially true for the mutation in the CXR #A6 clone, which resulted in the expression of a TOP2A lacking its terminal nuclear localization sequence and is therefore sequestered in the cytoplasm in interphase cells (*Figure 1—figure supplement 1C*), although it can access the DNA during mitosis. It is noteworthy that such a mutant would have been difficult to devise and express at proper levels in complementation experiments since TOP2A overexpression is toxic (*McPherson and Goldenberg, 1998*). Four of the eight amino acid changes in TOP2A identified in this work (P593S, S654N, L703I, and P890L) have been previously described and confer Vosaroxin resistance to complemented *Saccharomyces cerevisiae* (*Blower et al., 2019*). Vosaroxin is a quinolone derivative that acts as a DNA intercalator and a TOP2 inhibitor, further supporting the hypothesis that some of the TOP2A point mutations identified here alter TOP2A activity.

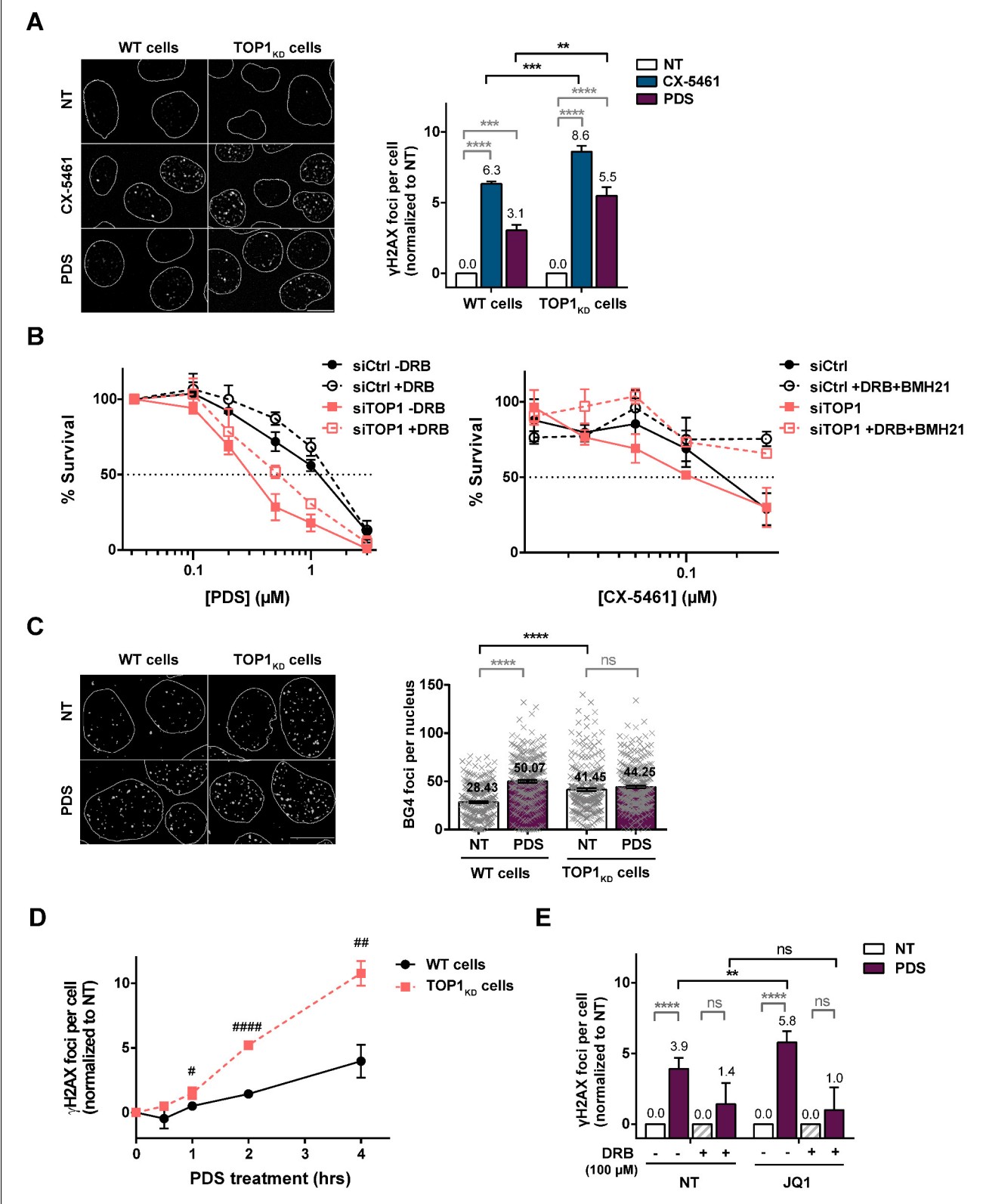

**Figure 6.** Topoisomerase 1 (TOP1) protein counteracts topoisomerase 2 (TOP2)-dependent double-stranded breaks (DSBs) induced by G-quadruplex (G4) stabilizers. (**A**) Representative images (left panel) and quantification (right panel) of γH2AX foci detected in HeLa control cells (wild-type [WT] cells) or HeLa TOP1$_{KD}$ (TOP1 knock-down cells by inducible shRNA-mediated depletion) and treated with pyridostatin (PDS) or CX-5461 for 4 hr. (**B**) Cell survival assay as assessed by clonogenic assay on HeLa cells transfected with control (Ctrl) or TOP1 siRNAs and treated with PDS +/- 5,6-

*Figure 6 continued on next page*

*Figure 6 continued*

dichloro-1-b-D-ribofuranosylbenzimidazole (DRB) (upper panel) or CX-5461 +/- DRB and BMH21 (bottom panel). For clonogenic assays PDS, CX-5461, DRB, and BMH21 treatments were performed as described in Materials and methods. (C) Representative images (left panel) and quantification (right panel) of BG4 foci fluorescence signal (gray) detected in HeLa control cells (WT cells) or HeLa TOP1$_{KD}$ cells treated with 20 µM PDS for 4 hr. (D) Kinetic studies of γH2AX foci formation in in HeLa control cells (WT cells) or TOP1 knock-down cells following PDS (20 µM) treatments. (E) Quantification of γH2AX signals in HeLa cells treated with PDS and the BRD4 inhibitor JQ1 in the presence of the RNA-Pol II inhibitor DRB. JQ1 was added 1 hr prior to addition of PDS. Expression of shTOPI in HeLa cells was induced with 5 µg/mL doxycycline for 5 days before treatments. Quantification of γH2AX foci per cell was performed on $n > 110$, $n > 181$, $n > 174$, and $n > 117$ nuclei for each condition, respectively, in (A), (C), (D), and (E). Error bars represent SD from the means, $n \geq 3$ independent experiments. p values were calculated using an unpaired multiple Student's $t$ test. ns: p>0.05; *p<0.05; **p<0.01; ***p<0.001; ****p<0.0001. Quantification of BG4 foci per cell was performed on $n > 71$ nuclei for each condition. Error bars represent SEM from the means, $n$ = three independent experiments. p values were calculated using an unpaired Welch's $t$ test. ns: p>0.05; *p<0.05; **p<0.01; ***p<0.001; ****p<0.0001.

The online version of this article includes the following source data and figure supplement(s) for figure 6:

**Source data 1.** Raw unedited image and uncropped figure of the blot of the western blot from *Figure 6*.
**Source data 2.** Raw unedited image and uncropped figure of the blot of the western-blot from *Figure 6*.
**Figure supplement 1.** Not normalized data from *Figure 6*.Topoisomerase 1 (TOP1) protein counteracts topoisomerase 2 (TOP2)-dependent double-stranded breaks (DSBs) induced by G-quadruplex (G4) stabilizers.
**Figure supplement 2.** Topoisomerase 1 (TOP1) protein counteracts topoisomerase 2 (TOP2)-dependent double-stranded breaks (DSBs) induced by G-quadruplex (G4) stabilizers.
**Figure supplement 3.** Topoisomerase 1 (TOP1) protein counteracts topoisomerase 2 (TOP2)-dependent double-stranded breaks (DSBs) induced by G-quadruplex (G4) stabilizers.

## Differential contribution of TOP2 proteins to cytotoxicity induced by PDS and CX-5461

A major finding of our work is the differential contribution of TOP2 isoforms to cytotoxicity induced by both clastogenic G4 ligands. Indeed, while *Bruno et al., 2020* have previously shown a major role for the TOP2A protein in the induction of DSBs upon CX-5461 treatment, the impact of the TOP2A activity on PDS-induced DSBs has not been clearly reported. Here, we show that *TOP2A* single-point mutations found in both CXR and F14R cells are sufficient to confer resistance to both CX-5461 and PDS. In addition, the major role of TOP2A in resistance to both G4 ligands was confirmed through an RNA silencing approach in HAP1 and HeLa cells. Although some slight differences in the response to G4 ligands were observed between HAP1 and HeLa cells, detailed analysis in HeLa cells of the relative contribution of the two TOP2 isoforms to DNA break production following G4 ligand treatments also supports a major role for TOP2A in the formation of DSBs by CX-5461 and PDS. Indeed, while TOP2A depletion strongly impacted on DSB production in response to PDS and CX-5461, TOP2B protein was only partially involved in DNA break production in response to CX-5461.

The main role of TOP2A in producing transcription-associated DNA breaks in response to PDS is at first intriguing since TOP2B is considered as the principal TOP2 responsible for the resolution of topological stress associated with transcription (*Pommier et al., 2016*; *Nitiss, 2009a*; *Madabhushi, 2018*; *Austin et al., 2018*). In contrast, TOP2A is believed to resolve mainly topological constraints associated with replication and chromosome segregation, in line with its increased expression during S and G2 phases (*Pommier et al., 2016*; *Nitiss, 2009a*; *Akimitsu et al., 2003*; *Carpenter and Porter, 2004*). However, some studies also implicate TOP2A in transcription (*Mondal and Parvin, 2001*; *Ray et al., 2013*; *Yu et al., 2017*) and its activity is required for maximal production of transcription-dependent DNA breaks induced by ETP, a potent poison of TOP2 (*Tammaro et al., 2013*). Moreover, genome-wide analysis of TOP2A cleavage sites shows a significant enrichment of TOP2A on highly transcribed loci (*Zhang et al., 2006*). Interestingly, elevated transcription levels have been shown to promote G4 formation that are favored by negative superhelicity caused by the progression of RNA-Pol complexes through DNA (*Zheng et al., 2017*).

## Transcription drives TOP2-dependent G4-ligand-induced DSBs

Another important finding of our work is the major impact of transcription on the formation of TOP2-mediated DNA breaks following G4 ligand treatment. In this study, we show that DRB treatment, which specifically blocks RNA-Pol II transcriptional elongation, completely abrogates DSB formation induced by PDS and significantly reduces DNA break production induced by CX-5461. Interestingly, we found that DNA breaks induced by CX-5461 are strongly reduced when both RNA-

Pol II and RNA-Pol I activities are inhibited, indicating an important contribution of rDNA transcription in the cellular response to this ligand. Consistent with our results, active transcription of rDNA repeats has been found to enhance the sensitivity of cells to CX-5461 and DNA damage production (*Son et al., 2020*). In cells, topological stresses induced by transcription are mainly resolved by the TOP1 protein (*Kim and Jinks-Robertson, 2017*). Here, we show evidence for a major role of the TOP1 protein in countering DNA break formation by CX-5461 and PDS. Consistent with our data, TOP1 depletion in yeast drives genomic instability at highly transcribed G4-forming sequences (*Yadav et al., 2014*; *Yadav et al., 2016*).

We also demonstrate that impairing RNA-Pol II progression by inhibiting the transcriptional elongation-promoting BRD4 protein increases the number of DSBs induced by PDS. Interestingly, the inhibition of BRD4 and TOP1 activities as well as the reduction of RNA-Pol elongation has been shown to promote R-loops (*Shivji et al., 2018*; *Tuduri et al., 2009*; *Zhang et al., 2017*; *Kim et al., 2019*; *El Hage et al., 2010*; *Li et al., 2015*), a DNA-RNA secondary structure associated with the formation of transcription-dependent DNA breaks (*Aguilera, 2002*; *Gaillard et al., 2013*; *Gómez-González and Aguilera, 2019*; *Aguilera and García-Muse, 2012*). In this study, we show that the expression of the RNaseH1 protein, which resolves R-loop structures, provokes a significant decrease of DSBs induced upon G4-ligand treatments. In human cells, G4-forming sequences are highly correlated with R-loop-forming regions (*Puget et al., 2019*; *Zheng et al., 2017*; *Zhang et al., 2020*); therefore, our results suggest that some of the DSBs induced upon G4 stabilization by G4 ligands are associated with R-loop formation. Consistent with our data, De Margis et al. have recently reported that DNA damage induced by G4 ligands in human cells is mediated through R-loop formation (*De Magis et al., 2019*).

## Why are G4 ligands toxic?

In cells, G4 ligands have been shown to affect cell growth through different mechanisms, altering telomere stability, replication, transcription, RNA metabolism, and mitochondrial maintenance (*Varshney et al., 2020*). Although cell alterations induced by these ligands have been linked to stabilization of G4 structures, their differential impact on different targets is probably related to their chemical structure, affecting their molecular activity and/or localization. For instance, the main impact of RHPS4 on mitochondrial DNA is related to its ability to translocate to the mitochondria, which is most probably due to its positive charge (*Falabella et al., 2019*). Our study, together with very recent reports (*Bruno et al., 2020*; *Olivieri et al., 2020*), indicates that cytotoxic effects induced by CX-5461 and PDS mainly rely on the formation of DNA breaks through a TOP2-dependent mechanism, arguing for the presence in these molecules of structural determinants that may block TOP2 activity during its active cycle. Supporting this assumption, competition experiments with 360A showed that stabilization of G4 structures is not sufficient to provoke TOP2-dependent DNA breaks. Of note, CX-5461, which was initially identified as a potent and selective RNA-Pol I inhibitor (*Haddach et al., 2012*), is a quarfloxin derivative, a small compound originally derived from the fluoroquinolone family that shows dual TOP2 inhibitor and G4-binding activities (*Brooks and Hurley, 2010*). Although the global TOP2 poisoning activity of quarfloxin derivatives has been shown to be attenuated by increased selectivity for G4 structures (*Brooks and Hurley, 2010*), we assume in our model that the TOP2 poisoning activity of CX-5461 is restricted to G4-forming regions.

In cells, inhibition of DNA-PKcs activity dramatically increases the number of DNA break signals in PDS and CX-5461-treated cells, demonstrating that an important number of G4 ligand-induced DNA breaks are repaired through the NHEJ pathway (*Xu et al., 2017*; *McLuckie et al., 2013*). This result suggests that most G4 ligand-induced breaks are efficiently repaired by NHEJ, while some TOP2-dependent DNA breaks refractory to repair probably cause the toxicity of these molecules.

On the basis of the results obtained in this study, and very recent reports from *Bruno et al., 2020*; *Olivieri et al., 2020*, we propose a model in which G4 ligands CX-5461 and PDS act as 'G4-dependent TOP2 poisons' (*Figure 7*). In this model, the interaction of both compounds with DNA is facilitated by DNA topological stress provoked by transcription. G4 stabilization by G4 ligands in transcriptionally active loci would provoke sustained RNA-Pol arrest, mobilizing topoisomerase enzymes to resolve topological stresses that at some loci may be poisoned in the vicinity of G4. Our model unifies the topoisomerase poisoning and G4-binding properties of these molecules in the new concept of DNA structure-driven topoisomerase poisoning at G-rich transcribed sequences.

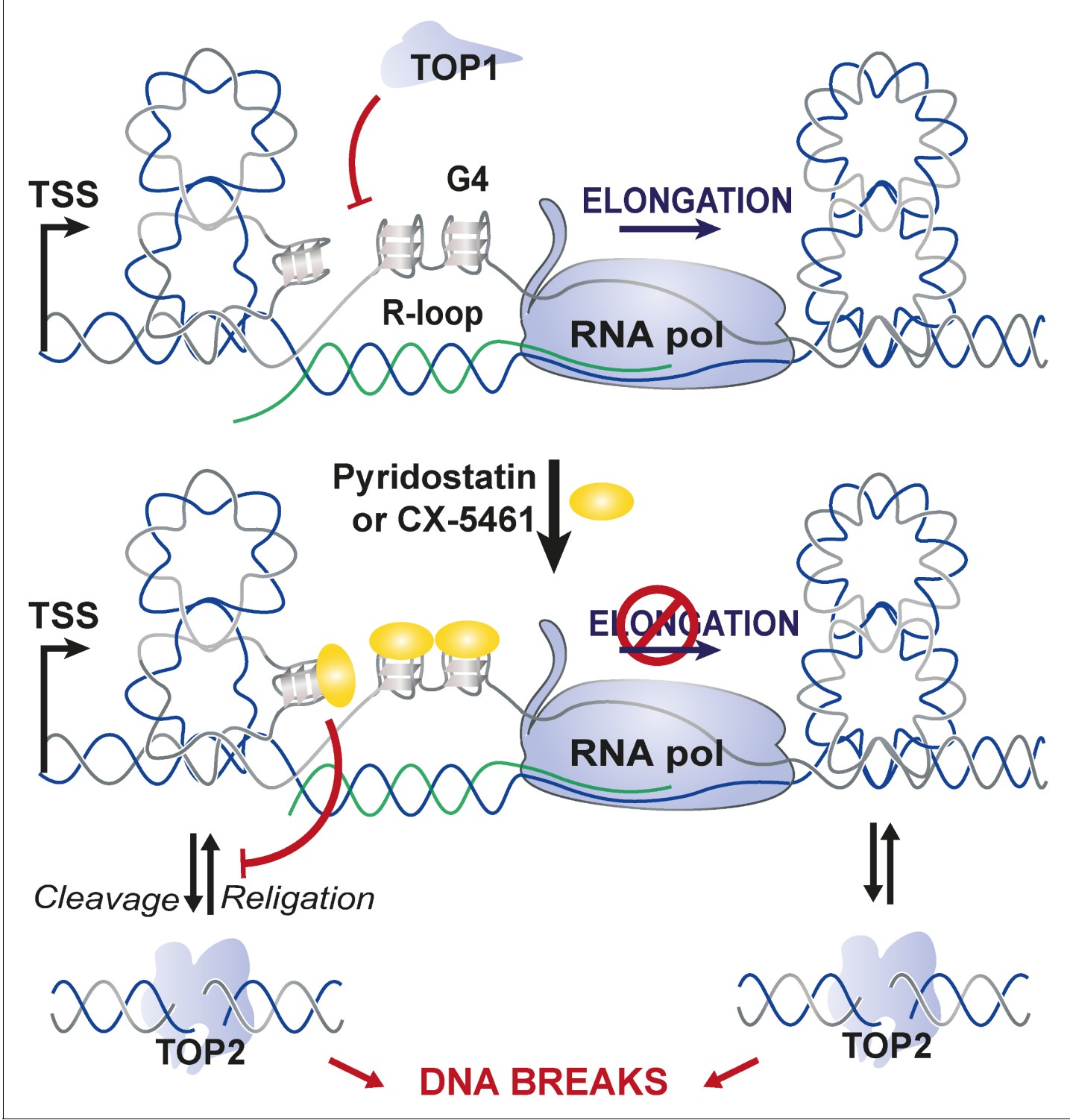

**Figure 7.** Proposed model for topoisomerase 2-mediated double-stranded breaks (DSBs) on transcriptionally active loci containing G-quadruplex (G4)-forming sequences. In this model, the interaction of G4 ligands with DNA is facilitated by DNA topological stress provoked by RNA-Pol-dependent transcription that are counteracted by topoisomerase 1 (TOP1) activity. G4 stabilization by G4 ligands in transcriptional active loci would provoke sustained RNA-Pol arrest mobilizing topoisomerase enzymes to resolve topological stresses and that at some loci may be poisoned at the vicinity of G4.

# Materials and methods

## Cell lines

All the cell lines used in this study were obtained from ATCC that certified their identity. DAPI staining analysis was used to confirm the absence of *Mycoplasma* contamination during the course of experiments.

## Cell culture conditions and treatments

All culture media were provided by Gibco and were supplemented with 10% fetal bovine serum (Eurobio), 100 U/mL penicillin (Gibco), and 100 µg/mL streptomycin (Gibco). Cells were grown in a humidified atmosphere with 5% $CO_2$ at 37°C. HeLa and U2OS cells were grown in Dulbecco's Modified Eagle Medium. HAP1 cells were cultured with Iscove's Modified Dulbecco's Medium. RPE1-hTERT cells were cultured with RPMI Media 1640 buffered with 0.3% $Na(CO_3)_2$. Expression of shTOPI in HeLa cells was induced with 5 µg/mL doxycycline for 5 days before treatments and for 4 days before siTOP2A transfections (described below) and maintained during siRNA-mediated protein depletion. RNaseHI-mCherry expression in U2OS cells was induced with 2.5 µg/mL doxycycline for 14 hr.

For selection of HAP1-resistant clones, haploid HAP1 were isolated using cell sorting and $100.10^6$ haploid HAP1 were mutagenized by treatment with 300 µg/mL EMS (Sigma-Aldrich) for 3 days. After a 1-week recovery, $0.5–1.10^6$ cells were seeded in 140 mm dishes. Plates were treated twice at a 1-week interval with 0.3 µM CX-5461 or 30 nM F14512 for 4 days. Around 10 days after the second treatment, individual clones (CXR and F14R) were isolated and used for further studies.

For immunofluorescence studies in U2OS, PDS was used at 20 µM for 8 hr. For immunofluorescence studies in HeLa and HAP1 cells, PDS and CX-5461 were used respectively at 20 µM and 0.2 µM for 4 hr. The G4 ligand 360A (20 µM) was added to cells 3 hr prior to PDS, CX-5461, and ETP (3.75 µM) treatments and renewed for the duration of the treatment. EdU treatments were performed at 10 µM at the same time as PDS and CX-5461 treatments and were maintained for the duration of the experiments. For inhibitors, DNA-PKi (2 µM, NU7441) and JQ1 (2 µM, (+/-)-JQ1) were added to cells 1 hr prior to PDS and or CX-5461 treatments. BNS22 (5 µM) was added to cells 30 min prior to PDS and or CX-5461 treatments. The transcription inhibitors DRB and BMH21 were used at 100 µM and 0.2 µM, respectively, and added to cells 1 hr prior to treatments. All inhibitors remained on cells for the duration of the experiment. For flow cytometry studies, HAP1 cells were incubated 1 hr with 10 µM EdU or exposed to 10 Gy X-ray irradiation (exposure time of 11 min 53 s) using a calibrated irradiation system (RX-650; Faxitron) fitted with a 0.5 mm aluminum filter for soft X-rays.

| Chemical compound | Furnisher | CAS number |
|---|---|---|
| BMH21, N-[2-(Dimethylamino)ethyl]−12-oxo-2H-benzo[g]pyrido[2,1-b]quinazoline-4-carboxamide | Sigma-Aldrich | 896705-16-1 |
| BNS-22 | Sigma-Aldrich | 1151668-24-4 |
| Calicheamicin γ1 was generously gifted by P.R. Hamann (Wyeth Research, Pearl River, NY, USA) | | |
| Camptothecin | Sigma-Aldrich | 7689-03-4 |
| CX-5461 | Selleckchem | 1138549-36-6 |
| DNA-PKi – NU7441 | Tocris | 503468-95-9 |
| Doxycycline hydrochloride | Sigma-Aldrich | 10592-13-9 |
| DRB, 5,6-dichloro-1-b-D-ribofuranosylbenzimidazole | Sigma-Aldrich | 54-85-0 |
| EdU, 5-Ethynyl-2'-deoxyuridine | Thermo Fisher Scientific | A10044 |
| Etoposide (ETP) | Sigma-Aldrich | 33419-42-0 |
| F14512 | Pierre Fabre Laboratories | 866874-63-7 |
| JQ1, (+/-)-JQ1 | Sigma-Aldrich | 1268524-69-1 |
| Nocodazole | Sigma-Aldrich | 31430-18-9 |
| PhenDC3 | Sigma-Aldrich | 929895-45-4 |

*Continued*

| Chemical compound | Furnisher | CAS number |
|---|---|---|
| Pyridostatin | Sigma-Aldrich | 1085412-37-8 |
| RHPS4 | Selleckchem | 390362-78-4 |

## Plasmid and cell constructions

U2OS cells conditionally expressing the RNaseH1-mCherry fusion protein were previously described in *Britton et al., 2014*. To construct HeLa cells conditionally expressing shRNAs against TOP1 mRNA, HeLa were infected with pLV-tTR-KRAB-Red and pLVTHM shTOP1 lentiviral particles. Individual clones from the transduced cell population were then isolated and selected for their capacity to downregulate TOP1 expression under treatment by the tetracycline analog doxycycline. pLV-tTR-KRAB-Red is a lentiviral vector encoding the transcriptional repressor tTR-KRAB fused to the DsRed fluorescent protein. pLVTHM is a lentiviral vector allowing conditional expression of an shRNA of interest under the control of the H1 promoter and the tetracycline operator/repressor system (TetO/TetR). pLVTHM vector allowing conditional expression of an shRNA against TOP1 was obtained by inserting duplex oligonucleotides (5'-CGCGTCCCCGGACTCCATCAGATACTATTTCAAGAGAATAG TATCTGATGGAGTCCTTTTTGGAAAT-3' and 5'- CGATTTCCAAAAAGGACTCCATCAGATACTATTC TCTTGAAATAGTATCTGATGGAGTCCGGGGA-3') between MluI and ClaI restriction sites in the pLVTHM plasmid. Transfection of HEK-293T cells (kindly provided by Genethon, Evry, France) with pLV-tTR-KRAB-Red or pLVTHM, and preparation of high titer lentiviruses pseudotyped with VSV-G protein have been performed as previously described (*Delenda, 2004*). pLV-tTRKRAB-red and pLVTHM were a gift from Didier Trono (Addgene plasmid # 12250; http://n2t.net/addgene:12250; RRID:Addgene_12250 and Addgene plasmid # 12247; http://n2t.net/addgene:12247; RRID:Addgene_12247) (*Wiznerowicz and Trono, 2003*).

## RNA interference

HeLa cells and HAP1 cells were respectively seeded at 250,000 cells and 400,000 cells per well in a 6-well plate. siRNAs oligonucleotides (see table below) were transfected twice (24 and 48 hr after seeding) at 50 nM final concentration per well with Lipofectamine RNAiMAX Reagent (Thermo Fisher Scientific) according to the manufacturer's recommendations. For TOP2A and TOP2B co-depletion, each siRNA was used at the final concentration of 25 nM. Cells were split 24 hr after the second-round transfection for immunodetection, immunoblotting, and viability assays, and were treated 24 hr after being seeded.

| Target | Name | Sequence or reference manufacturer | Manufacturer |
|---|---|---|---|
| Luciferase | siCtrl | 5'-CUUACGCUGAGUACUUCGATT-3' | Eurofins Genomics |
| TOP1 | siTOP1.2 | 5'-GGACUCCAUCAGAUACUAUUTT-3' | Eurofins Genomics |
| TOP2A | siTOP2A | 5'-CGUAGGCUGUUUAAAGAAATT-3' | Eurofins Genomics |
| TOP2A | siTOP2A5 | 5'-GAAUAACCAUAGAAAUGAATT-3' | Qiagen |
| TOP2A | siTOP2A7 | 5'-GCGUGGUCAAAGAGUCAUUTT-3' | Qiagen |
| TOP2B | siTOP2B | 5'-GGGCUAGGAAAGAAGUAAATT-3' | Qiagen |

## RNA-seq

RNA-seq was performed at the GeT-PlaGe core facility, INRA Toulouse, from total RNA prepared with the RNeasy Plus Mini Kit (Qiagen) according to the manufacturer's instructions. RNA-seq libraries were prepared according to Illumina's protocols using the Illumina TruSeq Stranded mRNA sample prep kit. Briefly, mRNAs were selected using poly-dT beads. Then, RNAs were fragmented and adaptors ligated. Eleven cycles of PCR were applied for library amplification. Library quality was assessed using a Fragment Analyser System (Agilent), and libraries were quantified by Q-PCR using

the Kapa Library Quantification Kit (Roche). RNA-seq experiments were performed on an Illumina HiSeq3000 using a paired-end read length of 2 × 150 bp.

## RNA-seq alignment and SNP prediction and filtering

Read quality was checked within the ng6 environment (*Mariette et al., 2012*) using fastQC (http://www.bioinformatics.babraham.ac.uk/projects/fastqc/) and Burrows-Wheeler Aligner (BWA) (*Li and Durbin, 2009*) to search for contamination. The reads were cleaned with cutadapt v1.8.3 and aligned against hg38 reference human genome with STAR v2.5.2b (*Dobin et al., 2013*). Expression levels were computed with featureCount (*Liao et al., 2014*) using Ensembl annotation. Alignments were deduplicated with samtools rmdup and reads not uniquely mapped removed. Then GATK v3.5 base quality score recalibration was applied (*McKenna et al., 2010*). Indel realignment, SNP, and INDEL discovery were performed with HaplotypeCaller using standard hard filtering parameters according to GATK Best Practices recommendations for RNA-seq. Finally, variants were annotated using snpEff v4.3T (*Cingolani et al., 2012*). A Python script was used to select protein coding variants specific to CXR clones as compared to wild-type HAP1, with a minimal allele frequency of 0.9 and a depth greater than 10 reads. Among these variants, we selected variants resulting in frameshifts, mis- and non-sense mutations as compared to the reference human genome hg38. Cytoscape v3.2.0 (*Shannon et al., 2003*) was used to identify genes found mutated in several CXR clones. Upon TOP2A identification as a common gene mutated in five CXR clones, IGV v2.4.15 was used to scrutinize alignment data and revealed two TOP2A mutations missed by the analysis: for CXR #A2, the point mutation S654N; and, for CXR #A6, a mutation of the first nucleotide of the last intron leading to intron retention. Clone clustering under Cytoscape, based on shared mutated genes, suggested a common origin for clones CXR #A1, #A3, #A5, and #B4 (multiple common mutations). RNA-seq data from wild-type HAP1 and CXR clones have been deposited on SRA with the project ID PRJNA637883.

## Targeted sequencing of TOP2A cDNA from HAP1 clones

Total RNAs were extracted from wild-type or F14R HAP1 with the RNeasy Plus Mini Kit (Qiagen) according to the manufacturer's instructions. TOP2A cDNA was produced from these RNAs with the Superscript III First-Strand kit (Thermo Fisher Scientific) according to the manufacturer's instructions and using the TOP2A-Rv primer. The resulting TOP2A cDNAs was amplified in four overlapping fragments using the primer pairs [TOP2A-F1, TOP2A-R1], [TOP2A-F2, TOP2A-R2], [TOP2A-F3, TOP2A-R3], and [TOP2A-F4, TOP2A-Rv] and sequenced using the same primers except for the last fragment for which the TOP2A-R4 sequencing primer was also used.

| Name | Oligonucleotide sequence (5′ to 3′) |
| --- | --- |
| TOP2A-F1 | GTCGCTTTCAGGGTTCTTGAGCC |
| TOP2A-R1 | TGGCATGTTGATCCAAAGCTCTTGG |
| TOP2A-F2 | TGGTGTTGCAGTAAAAGCACATCAGG |
| TOP2A-R2 | GCAACCTTTACTTCTCGCTTGTCATTCC |
| TOP2A-F3 | TCCTGAGGATTACTTGTATGGACAAACTACC |
| TOP2A-R3 | GCCTTCACAGGATCCGAATCATATCCC |
| TOP2A-F4 | GGCTCCTAGGAATGCTTGGTGC |
| TOP2A-R4 | TCATCTGGGAAATGTGTAGCAGGAGG |
| TOP2A-Rv | GCTTCAGGTAACTTTAAAACCAGTCTTGG |

## Heparin-based extraction for TOP2cc immunodetection

This method was adapted from *de Campos-Nebel et al., 2016*. For immunofluorescence, HeLa cells were seeded as described below. Cells were washed with ice-cold PBS and incubated twice for 5 min on ice with CSK buffer (10 mM PIPES pH 7.0, 100 mM NaCl, 300 mM sucrose, 3 mM MgCl₂) containing 0.7% Triton X-100 and 20 U/mL Heparin (Sigma-Aldrich). Cells were washed with ice-cold PBS and fixed on ice 15 min with paraformaldehyde 2% in PBS. Then, immunofluorescence was

performed as described below without the permeabilization step. For immunoblotting, HAP1 cells were seeded at $1.5 \cdot 10^6$ cells in 6 cm dishes 24 hr prior ETP treatment (200 µM, 1 hr). After treatment, cells were harvested with trypsin and washed with cold PBS. After gentle centrifugation, cells were resuspended in lysis buffer (150 mM NaCl, 1 mM EDTA, 0.5% IGEPAL CA-630, 2X HALT Protease and Phosphatase Inhibitor Cocktail [Thermo Fisher Scientific] 20 mM Tris-HCl, pH 8.0) complemented with 100 U/mL Heparin (H3393, Sigma-Aldrich) and incubated on ice for 15 min. Then lysates were centrifugated at 15,000 rpm at 4°C for 5 min and pellets were resuspended with lysis buffer. In order to facilitate migration on polyacrylamide gel, a sonication was performed to degrade DNA present within the extracts. Protein concentrations were determined by measuring absorbance at 280 nm (Nanodrop) and heparin-based extracts were diluted with denaturing lysis buffer (120 mM Tris-HCl pH 6.8, 4% SDS, 20% glycerol). Western blot was performed as described below.

## Cell lysis and western blotting

Whole-cell extracts were prepared from PBS-washed pellet lysed with denaturing lysis buffer (120 mM Tris-HCl pH 6.8, 4% SDS, 20% glycerol) and 10 strokes through a 24G needle. Protein concentrations were determined by measuring absorbance at 280 nm (Nanodrop). For loading, an equal volume of a solution of 0.01% bromophenol blue and 200 mM dithiothreitol was added to the extracts, then boiled at 95°C for 5 min. About 80 µg of denatured proteins were loaded for each condition and separated on standard or Stain-Free gradient 4–12% polyacrylamide TGX pre-cast gels (Bio-Rad) and transferred onto nitrocellulose membranes (0.45 µm pore size, Bio-Rad or Protran, GE Healthcare). Before blocking (incubation 1 hr at room temperature in PBS 0.1% Tween-20, 5% non-fat dry milk), Ponceau S staining or UV exposition of membrane (for Stain-Free gels) was used to confirm homogeneous loading. The membrane was successively probed with primary antibody and appropriate goat secondary antibodies coupled to horseradish peroxidase (described in table below). A ChemiDoc CCD imager (Bio-Rad) was used to acquire pictures of the stain-free total protein staining and the chemiluminescence signal after membrane incubation with adequate peroxidase substrate (Clarity ECL, Bio-Rad). Digital data were processed and quantified using ImageJ software.

| Target | Dilution | Species | Class | Reference | Manufacturer |
|---|---|---|---|---|---|
| αTubulin | 1:25,000 | Mouse | Monoclonal | T5168 | Sigma-Aldrich |
| KU80 (*Dobin et al., 2013*) | 0.2 µg/mL | Mouse | Monoclonal | MA5-12933 | Thermo Fisher Scientific |
| KU70 (N3H10) | 0.2 µg/mL | Mouse | Monoclonal | MA5-13110 | Thermo Fisher Scientific |
| TOP1 | 1:1000 | Rabbit | Monoclonal | EPR5375 | Abcam |
| TOP2A (aa1352-1493) | 1 µg/mL | Mouse | Monoclonal | GTX35137 | Genetex |
| TOP2A (aa1500-C-term) | 1 µg/mL | Rabbit | Polyclonal | A300-054B | Bethyl Laboratories |
| TOP2B | 0.2 µg/mL | Rabbit | Polyclonal | A300-950A | Bethyl Laboratories |
| Anti-rabbit HRP-coupled | 1:10,000 | Goat | Polyclonal | 111-035-003 | Jackson Immunoresearch |
| Anti-mouse HRP-coupled | 1:10,000 | Goat | Polyclonal | 115-035-003 | Jackson Immunoresearch |

## Immunofluorescence

RPE1-hTERT cells were seeded, treated, and stained in the same conditions than HeLa cells. HeLa cells and HAP1 cells were seeded in 24-wells plate at respectively 100,000 cells/well and 25,000 cells/well on #1.5 glass coverslips (VWR, #631-0150). HeLa cells and HAP1 cells were respectively treated 24 hr and 48 hr later, and then fixed with paraformaldehyde 2% in PBS at room temperature (10 min for HeLa cells, 15 min for HAP1 cells), washed with PBS, and permeabilized for 15 min at room temperature with 10 mM Tris-HCl pH 7.5, 120 mM KCl, 20 mM NaCl, 0.1% Triton-X 100. In EdU- treated cells, cells were washed with PBS and EdU detection reaction was performed with Click-iT RNA Alexa Fluor Imaging Kit according to the manufacturer's recommendation, with 2 µM Alexa Fluor 594 or 648 azide for 30 min at room temperature. Then, cells were washed with PBS and incubated for about 1 hr at room temperature in blocking buffer (20 mM Tris-HCl pH 7.5, 150 mM NaCl, 2% BSA, 0.2% fish gelatin, 0.1% Triton-X 100) prior to incubation overnight at 4°C with primary

antibody diluted in blocking buffer (dilutions shown in table below). For BG4 immunodetection, blocking buffer were complemented 0.3 µg/µL of RNAse A (*David et al., 2019*). Cells were then washed with PBS 0.1% Tween-20 and incubated with appropriate secondary goat antibody coupled to Alexa Fluor 488 or 594 diluted in blocking buffer (dilutions shown in table below) for 1 hr at room temperature. For TOP2Acc and BG4 co-immunodetection, both primary anti-BG4 antibody and appropriate secondary antibody were successively added to cells prior to incubations to primary anti-TOP2A antibody and its appropriate secondary antibody. At last, cells were washed with PBS 0.1% Tween-20 and stained with 0.1 µg/mL DAPI for 20 min at room temperature, and coverslips were mounted with Vectashield mounting medium (Vector Laboratories). Nuclear γH2AX foci, 53BP1 foci, BG4 foci, TOP2Acc foci, and EdU-integrated density staining overlapping with DAPI staining were quantified with ImageJ software. Nuclear DAPI-integrated density staining was quantified with ImageJ software and correlated to nuclear EdU-integrated density to determine cell cycle phase for each cell as described in *Roukos et al., 2015*. Quantifications of nuclear γH2AX, 53BP1, and TOP2Acc foci induced by G4 ligands or ETP are represented normalized to non-treated (NT) conditions. For TOP2Acc imaging, images were acquired with a Zeiss Elyra 7 3D Lattice SIM super-resolution microscope fitted with a 63× objective (PLANAPO NA 1.4, Zeiss) and dual sCMOS cameras (pco.edge). 3D-SIM reconstructions were performed with Zen Black 2.3 (Zeiss). For TOP2Acc and BG4 co-immunodetection, images were obtained by performing a maximum intensity projection of 20 3D-SIM Z-stacks (interval 0.091 µM) with Zen Blue 3.3 (Zeiss). Quantification of foci and colocalization events was done manually. Zen Blue 3.3 was used to adjust brightness and contrast of corresponding micrographs as well as image cropping.

| Target | Dilution (µg/mL) | Species | Class | Reference | Manufacturer |
|---|---|---|---|---|---|
| 53BP1 | 1.3 | Mouse | Monoclonal | MAB-3803 | Millipore |
| γH2AX (Phospho S139) | 0.7 | Rabbit | Monoclonal | 81299 | Abcam |
| BG4 | 0.25 | Mouse | Monoclonal | Ab00174-1.1 | Absolute antibody |
| TOP2A | 0.5 | Rabbit | Polyclonal | A300-054B | Bethyl Laboratories |
| TOP2A | 0.2 | Mouse | Monoclonal | GTX35137 | Genetex |
| Anti-rabbit 488 | 2 | Goat | Polyclonal | A11008 | Thermo Fisher Scientific |
| Anti-mouse 594 | 2 | Goat | Polyclonal | A11005 | Thermo Fisher Scientific |
| Anti-mouse 488 | 2 | Goat | Polyclonal | A11001 | Thermo Fisher Scientific |
| Anti-rabbit 594 | 2 | Goat | Polyclonal | A11012 | Thermo Fisher Scientific |

## Viability assay (SRB)

HAP1 cells were seeded in 96-flat-wells plate at 3500 cells per well. Serial dilutions of various compounds were realized allowing same solvent concentration for each condition, and cells were treated 24 hr after seeding. After 3 days, HAP1 cells were fixed for 1 hr at 4°C by addition of 10% trichloroacetic acid to a 3.33% final concentration, before being washed with tap water and dried overnight. Cells were stained by incubation 30 min at room temperature in a 1% acetic acid solution containing 0.057% sulforhodamin B, then cells were washed with 1% acetic acid and dried overnight. Finally, 200 µL of a 10 mM Tris-base solution was added, plates were agitated for 1 hr at room temperature, and SRB levels were measured by absorbance at 490 nm using µQuant microplate spectrophotometer (Bio-Tek Instruments). Percentages of cell viability are expressed after normalization relative to NT controls. For characterization of wild-type and drug-resistant HAP1, the $IC_{50}$ (50% inhibitory concentration) was computed for each drug and cell lines with the GraphPad Prism v8 software using a nonlinear regression to a four-parameter logistic curve.

## Cell proliferation assay

HAP1 cells were seeded in 6-well plates at 40,000 cells/well. The occupied area (% confluency) was monitored every hour for 74 hr using a live-imaging Incucyte Zoom system (Sartorius). Analysis of the 7–72 hr exponential part of the occupied area vs. time exponential curve was used to compute the doubling time of wild-type and resistant HAP1 cells using GraphPad Prism v8. The analysis was

repeated three times, and the scatter dot-blot shows the three values, the mean, and the SD from the mean. A one-way ANOVA test was used and revealed no significant difference between the doubling time of resistant clones and the WT HAP1.

## Clonogenic assay

Clonogenic assay was performed as described by *Bombarde et al., 2017*. Briefly, after transfection with siRNA, HeLa cells was seeded at low density (250 cells/well) the day before treatment, pre-incubated with 100 μM DRB, 0.2 μM BMH21, or dimethylsulfoxide (DMSO) for 1 hr and treated for 4 hr with PDS or CX-5461 in the presence of transcription inhibitor or DMSO before being replaced by fresh medium. After 10–15 days, cells were stained with crystal violet and the colonies were counted (at least 50 colonies were counted for each condition per experiment). Data were normalized to the NT conditions. The $IC_{50}$ was computed in each condition with the GraphPad Prism v8 software using a nonlinear regression to a four-parameter logistic curve.

## Flow cytometry

Briefly, HAP1 cells were collected by trypsination at the end of the treatment, washed with PBS 1% BSA, and fixed for 15 min with paraformaldehyde 2% in PBS at room temperature. Cells were washed with PBS 1% BSA, incubated for 30 min with PBS 0.2% Triton X-100, and washed again with PBS 1% BSA. If necessary, EdU detection reaction was performed as described above. If not, cells were incubated for 1 hr with primary anti-γH2AX antibody diluted in PBS 0.1% Tween-20 5% BSA (1:1000, ref ZMS05636, Sigma-Aldrich), washed with PBS 1% BSA, and incubated 30 min with PBS 0.1% Tween-20 5% BSA containing appropriate secondary antibodies coupled to Alexa Fluor 488 (1:200). Lastly, cells were washed with PBS 1% BSA and incubated in PBS containing 250 μg/mL RNAse A and 2 μg/mL DAPI. A BD LSR II flow cytometer (Becton Dickinson) was used to analyze a minimum of 30,000 cells. Data were analyzed and formatted using FlowJo v8.8.7.

## Statistical analyses

All results represent at least three independent experiments. Statistical analyses were performed with the GraphPad Prism software (version 8). For γH2AX and 53BP1 quantifications analyses, multiple unpaired *t* tests (without corrections for multiple comparisons) were performed between pairs of conditions. For BG4 quantifications analyses, results of at least three independent experiments were pulled together and unpaired Welch's *t* tests were performed between pairs of conditions. On all figures, significant differences between specified pairs of conditions are shown by asterisks (*p-value<0.05; **p-value<0.01; ***p-value<0.001; ****p-value<0.0001). NS is for nonsignificant difference.

## Acknowledgements

This work was funded by grants from ANR (ANR-17-CE18-0002-01 and ANR-16-CE11-0006-01), Cancéropôle GSO Emergence funding 'CX-Break,' and La Ligue Nationale Contre le Cancer (Equipe Labellisée 2018). We thank Julia Coates and Steve Coates for proofreading the manuscript. We are grateful to Emmanuelle Näser, Antonio Peixoto, and Elodie Vega for technical assistance and the Imaging Core Facility TRI-IPBS, supported by ITMO Cancer (Alliance Nationale pour les Sciences de la Vie et de la Santé, National Alliance for Life Sciences and Health) within the Framework of the Cancer Plan. We thank P Mailliet for providing 360A. We also thank J Coates and S Coates for the careful proofreading and language editing of the manuscript. We thank 'Région Midi-Pyrénées' for supporting the 'Toulouse Réseaux Imagerie' platform. Patrick Calsou is a researcher from INSERM. This work was performed in collaboration with the GeT core facility, Toulouse, France (http://get.genotoul.fr), and was supported by France Génomique National infrastructure, funded as part of 'Investissement d'avenir' program managed by Agence Nationale pour la Recherche (contract ANR-10-INBS-09).

## Additional information

### Funding

| Funder | Grant reference number | Author |
|---|---|---|
| Agence Nationale de la Recherche | ANR-17-CE18-0002-01 | Madeleine Bossaert<br>Linh-Trang Nguyên<br>Patrick Calsou<br>Sébastien Britton<br>Dennis Gomez |
| Agence Nationale de la Recherche | ANR-16-CE11-0006-01 | Angélique Pipier<br>Jean-Francois Riou<br>Eric Defrancq<br>Patrick Calsou<br>Sébastien Britton<br>Dennis Gomez |
| Cancéropôle Grand Ouest | Emergence funding "CX-Break" | Madeleine Bossaert<br>Angélique Pipier<br>Linh-Trang Nguyên<br>Patrick Calsou<br>Sébastien Britton<br>Dennis Gomez |
| Ligue Contre le Cancer | Equipe Labellisée 2018 | Madeleine Bossaert<br>Angélique Pipier<br>Linh-Trang Nguyên<br>Patrick Calsou<br>Sébastien Britton<br>Dennis Gomez |
| Agence Nationale de la Recherche | ANR-10-INBS-09 | Céline Noirot<br>Remy-Felix Serre<br>Olivier Bouchez |

The funders had no role in study design, data collection and interpretation, or the decision to submit the work for publication.

### Author contributions

Madeleine Bossaert, Angélique Pipier, Formal analysis, Investigation; Jean-Francois Riou, Writing - original draft; Céline Noirot, Formal analysis; Linh-Trang Nguyên, Remy-Felix Serre, Olivier Bouchez, Investigation; Eric Defrancq, Resources, Writing - original draft; Patrick Calsou, Conceptualization, Writing - original draft; Sébastien Britton, Conceptualization, Supervision, Investigation, Writing - original draft; Dennis Gomez, Conceptualization, Formal analysis, Supervision, Investigation, Writing - original draft

### Author ORCIDs

Madeleine Bossaert https://orcid.org/0000-0002-8792-6783
Angélique Pipier http://orcid.org/0000-0002-8481-2860
Jean-Francois Riou https://orcid.org/0000-0002-0055-6506
Sébastien Britton https://orcid.org/0000-0002-7008-5316
Dennis Gomez https://orcid.org/0000-0001-9942-1451

### Decision letter and Author response

Decision letter https://doi.org/10.7554/eLife.65184.sa1
Author response https://doi.org/10.7554/eLife.65184.sa2

## Additional files

### Supplementary files

• Source data 1. Raw unedited images and uncropped figures of western-blot analyses.

• Supplementary file 1. $IC_{50}$ values as assessed by cell survival assays to CX-5461, pyridostatin (PDS), etoposide, F14512, and nocodazole, of wild-type (WT), CX-5461-resistant (CXR), and F14512-resistant (F14R) HAP1 cells. Topoisomerase 2α (TOP2A) mutation present in CXR and F14R clones are indicated in the first column.

• Supplementary file 2. Non- or mis-sense mutations found in CX-5461-resistant (CXR) clones. Nucleotide changes and resulting amino acid modifications are indicated for each mutated gene. [1]Topoisomerase 2α (TOP2A) mutations found by a manual analysis of RNA-seq data.

• Transparent reporting form

## Data availability

RNA-seq data from wild-type HAP1 and CXR clones have been deposited on SRA with the project ID PRJNA637883.

The following dataset was generated:

| Author(s) | Year | Dataset title | Dataset URL | Database and Identifier |
|---|---|---|---|---|
| Pipier Al, Bossaert M, Riou JF, Noirot Cl, Nguyên LT, Serre RF, Bouchez O, Defrancq E, Calsou P, Britton Sb | 2020 | RNA-seq sequencing data from individual CX-5461-resistant HAP-1 clones | https://www.ncbi.nlm.nih.gov/bioproject/PRJNA637883 | NCBI BioProject, PRJNA637883 |

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
