## [Decision Letter]

**Acceptance summary:**

The authors search for mutations in HAP1 haploid cells that lead to resistance to G4-binding drugs and find a number of mutations in TOP2 alpha. Despite the actual belief that Top2alpha is preferentially associated with replication, whereas TOP2 beta has a function related with transcription, the study provides evidence that TOP2 alpha mediates cytotoxicity of G4-ligands PDS and CX-5461 and that transcription is key for the formation of DNA breaks induced by these drugs at G4 regions. The study is relevant to understand genotoxic resistance of specific tumor cells.

**Decision letter after peer review:**

Thank you for submitting your article "Transcription-associated topoisomerase 2α activity is a major effector of cytotoxicity induced by G-quadruplex ligands" for consideration by *eLife*. Your article has been reviewed by 3 peer reviewers, one of whom is a member of our Board of Reviewing Editors, and the evaluation has been overseen by Jessica Tyler as the Senior Editor. The following individual involved in review of your submission has agreed to reveal their identity: Laurence H Hurley (Reviewer #3).

The reviewers have discussed the reviews with one another and the Reviewing Editor has drafted this decision to help you prepare a revised submission.

Summary:

In this study the authors search for mutations in HAP1 haploid cells that lead to resistance to one of these drugs, CX-5461. They identify 8 different mutations within Top2alpha that suppress the cytotoxic effect of G4 stabilization by G4 ligands. The study is then focused in testing how TOP2A deficiency lead to increase in DNA breaks as inferred from gammaH2AX foci analysis in different cell types after treatment with CX-5461 and PDS G4-binding drugs. Interestingly, despite the actual believe that Top2alpha is preferentially associated with replication, whereas TOP2beta has a function related with transcription, this study shows that the breaks mediated by TOP2alpha is replication-independent but transcription dependent. In addition, authors show that RNH1 suppress the phenotype suggesting that R loops are mediating the phenomenon. After knocking down TOP1 the authors show similar type of transcription-dependent accumulation of damage. Thus, while TOP 2 has been previously identified as the target for CX-5461 in recent papers published in well-respected journals (PNAS, and Cell) this paper provides further insight into the precise role of TOP2A by using an unbiased genetic approach. This paper builds upon what was previously known and importantly provides insight into drug combination that might be useful in the clinic. It is very elegant and nicely written. However, few additional experiments would be required to fill several holes that would strengthen the conclusions provided.

Essential revisions:

– Preferably a DRIP analysis or at least an IF detection of DNA-RNA hybrids should be provided minimally for cells treated with PDS to complete data in Figure 4

– A prediction of this study is that Top2 inhibition should suppress the increase of gammaH2AX observed in TOP1-KD cells as determined in Figure 5D. This needs to be tested using BNS22 in TOP1-KD cells

– Would authors expect a larger accumulation at G4-rich regions. Could authors check this possibility by checking TOP2cc binding sites at some G4-rich region using etoposide?

– The authors describe that Top2 releases stress at G4 which causes cytotoxicity. Is this also true at naturally forming G4s that form in cells for example due to mutations in helicases (e.g. WRN, BLM?)

– Why are the newly identified Top2 mutant cells happy? Is the expression of cell cycle checkpoints, DNA repair genes altered in their studies which could explain their observations?

– Other G4 ligands have been shown to cause reduced cellular growth by changing the expression of apoptotic and autophagic genes, or by causes DNA damage only at telomeres. Are here also Top2 mutations effecting the cytotoxicity?

– G4s in R-Loops are proposed to be the major target based upon reduction of DSBs in experiments in which RNaseH1 is expressed. These experiments are only illustrated with PDS but not CX-5461. It should also be shown for PDS.

---

## [Author Response]

Essential revisions:– Preferably a DRIP analysis or at least an IF detection of DNA-RNA hybrids should be provided minimally for cells treated with PDS to complete data in Figure 4

The impact of PDS and CX-5461 on R-loop formation in human cells is now supported by several recent studies from different groups (1-4), however our attempts to visualize R-loops by imaging with the S9.6 antibody were unsuccessful, giving variable results. A study from the Capranico’s lab support that PDS quickly induces R-loops, but that these structures are transient (1), which might explain our difficulties. However, to consolidate our findings, we now show in our revised manuscript that the expression of RNaseH1 inhibits DSBs formation by both CX-5461 and PDS treatments. These data strongly support that DSB induction by these two G4 ligands relies on the formation of R-loops.

– A prediction of this study is that Top2 inhibition should suppress the increase of gammaH2AX observed in TOP1-KD cells as determined in Figure 5D. This needs to be tested using BNS22 in TOP1-KD cells

The effect of BNS22 in the formation of gH2AX signals following CX-5461 and PDS treatments was evaluated and these new results are added in the revised manuscript (Figure 6-supplement 2A). These data show that the increase level of DSBs observed in TOP1 knock-down cells treated by G4 ligands is blocked by TOP2 inhibition, supporting the role of TOP1 as a key modulator of TOP2-dependent formation of DSBs upon CX-5461 and PDS treatments.

– Would authors expect a larger accumulation at G4-rich regions. Could authors check this possibility by checking TOP2cc binding sites at some G4-rich region using etoposide?

In the revised version of our manuscript (new main Figure 4) we evaluated through both competition experiments with a non clastogenic G4 ligand and through colocalization studies the association of TOP2Acc induced by CX-5461 and PDS with G4 structures in cells. To perform the later experiment, we developed and validated a novel approach to visualize and quantify TOP2Acc by immunofluorescence. The data presented in the new Figure 4 strongly support that CX-5461 and PDS induce TOP2Acc which are associated to G4 structures in cells. We anticipate that our immunofluorescence approach for TOP2cc imaging will arouse the interest of a large number of researchers.

– The authors describe that Top2 releases stress at G4 which causes cytotoxicity. Is this also true at naturally forming G4s that form in cells for example due to mutations in helicases (e.g. WRN, BLM?)

In our manuscript we described the role of TOP2 proteins in the formation of DSBs induced by two clastogenic G4 ligands CX-5461 and PDS. Unlike many other G4 ligands, CX-5461 and PDS have been shown to provoke a rapid accumulation of DNA breaks, indicating that both compounds act through particular molecular mechanisms that are not activated by other G4 ligands. Of note, as discussed in the revised version of our manuscript, CX-5461 is a quarfloxin derivative, a small compound showing a dual TOP2 and G4 interaction. Altogether these results argue that TOP2-dependent DSBs induced by CX-5461 and PDS rely most probably on structural motifs present in CX-5461 and PDS interfering with TOP2 activity in addition to their ability to stabilize G4 structures. In agreement with this hypothesis, competition experiments with the non clastogenic G4 binder 360A showed a strong inhibition of both TOP2Acc and gH2AX formation induced by subsequent CX-5461 and PDS treatments, supporting that the stabilization of “natural” G4 structures is not sufficient to provoke TOP2 mediated DSBs. Of course, while we cannot exclude that TOP2 proteins may induce breaks associated with “natural” and “permanent” G4 structures formed in particular context (e.g. mutations in DNA helicases), their number is most likely very low, as even the stabilization of G4 structures by CX-5461 and PDS only induces a few gH2AX foci. Thus, to investigate the impact of TOP2 in the formation of DSBs associated with “natural” G4 forming sequences, genomic approaches have now to be considered and optimized. We believe that such experiments cannot be perform upon reasonable delay and are beyond the scope of this manuscript that focus on G4 structures stabilized by G4 ligands.

– Why are the newly identified Top2 mutant cells happy? Is the expression of cell cycle checkpoints, DNA repair genes altered in their studies which could explain their observations?

To address this point, we performed multiple complementary experiments now displayed in Figure 2-supplement 2. First, we assessed the impact of TOP2A mutations on cell proliferation by comparing the population doubling-time and the cell cycle distribution, by flow cytometry and EdU staining, between the WT and mutant cells. Then, we analyzed the impact of TOP2A mutations on the response to DNA damage. For this we compared the sensitivity of WT and mutant cells to camptothecin and calicheamicin, two DNA damaging agents that do not act through G4. We also compared the reversal of gH2AX signal, as a readout of DNA repair, of WT and mutant cells after treatment with ionizing radiation. Finally, since we show in our work that DNA topoisomerase I (TOP1) counteracts the toxicity of G4 ligands, we also compared the expression level of TOP1 between the WT and mutant cells. All these experiments revealed no significant difference between the WT and mutant cells, indicating that the reduced level of TOP2A activity in the HAP1 clones is sufficient for the cells to remain healthy.

– Other G4 ligands have been shown to cause reduced cellular growth by changing the expression of apoptotic and autophagic genes, or by causes DNA damage only at telomeres. Are here also Top2 mutations effecting the cytotoxicity?

Related to the first question, we also analyzed, via survival assays in the Figure 2-supplement 2, the response of mutant cells to RHPS4 and PhenDC3, two chemically unrelated G4 ligands shown to affect cell growth through mechanisms that do not involve DSB formation. In this figure, we showed that TOP2A mutations do not modify the sensitivity to RHPS4 and PhenDC3, indicating that the TOP2A mutations confer a specific resistance to the clastogenic CX-5461 and PDS compounds.

– G4s in R-Loops are proposed to be the major target based upon reduction of DSBs in experiments in which RNaseH1 is expressed. These experiments are only illustrated with PDS but not CX-5461. It should also be shown for PDS.

The effect of RNaseH1 expression on the formation of DSBs following both CX-5461 and PDS treatments has been added to the revised version of our manuscript (Figure 5E).

References

1. De Magis, A., Manzo, S.G., Russo, M., Marinello, J., Morigi, R., Sordet, O. and Capranico, G. (2019) DNA damage and genome instability by G-quadruplex ligands are mediated by R loops in human cancer cells. Proc Natl Acad Sci U S A, 116, 816-825.

2. Sanij, E., Hannan, K.M., Xuan, J., Yan, S., Ahern, J.E., Trigos, A.S., Brajanovski, N., Son, J., Chan, K.T., Kondrashova, O. et al. (2020) CX-5461 activates the DNA damage response and demonstrates therapeutic efficacy in high-grade serous ovarian cancer. Nat Commun, 11, 2641.

3. Shen, W., Sun, H., De Hoyos, C.L., Bailey, J.K., Liang, X.H. and Crooke, S.T. (2017) Dynamic nucleoplasmic and nucleolar localization of mammalian RNase H1 in response to RNAP I transcriptional R-loops. Nucleic Acids Res, 45, 10672-10692.

4. Duquette, M.L., Handa, P., Vincent, J.A., Taylor, A.F. and Maizels, N. (2004) Intracellular transcription of G-rich DNAs induces formation of G-loops, novel structures containing G4 DNA. Genes Dev, 18, 1618-1629.